# Single-molecule imaging of chromatin remodelers reveals role of ATPase in promoting fast kinetics of target search and dissociation from chromatin

Jee Min Kim[1], Pat Visanpattanasin[1†], Vivian Jou[1†], Sheng Liu[1], Xiaona Tang[1], Qinsi Zheng[2], Kai Yu Li[1], Jonathan Snedeker[1], Luke D Lavis[2], Timothee Lionnet[3], Carl Wu[1,4]*

[1]Department of Biology, Johns Hopkins University, Baltimore, United States; [2]Janelia Research Campus, Howard Hughes Medical Institute, Ashburn, United States; [3]Institute of Systems Genetics, Langone Medical Center, New York University, New York, United States; [4]Department of Molecular Biology and Genetics, Johns Hopkins School of Medicine, Baltimore, United States

*For correspondence:
wuc@jhu.edu

†These authors contributed equally to this work

**Abstract** Conserved ATP-dependent chromatin remodelers establish and maintain genome-wide chromatin architectures of regulatory DNA during cellular lifespan, but the temporal interactions between remodelers and chromatin targets have been obscure. We performed live-cell single-molecule tracking for RSC, SWI/SNF, CHD1, ISW1, ISW2, and INO80 remodeling complexes in budding yeast and detected hyperkinetic behaviors for chromatin-bound molecules that frequently transition to the free state for all complexes. Chromatin-bound remodelers display notably higher diffusion than nucleosomal histones, and strikingly fast dissociation kinetics with 4–7 s mean residence times. These enhanced dynamics require ATP binding or hydrolysis by the catalytic ATPase, uncovering an additional function to its established role in nucleosome remodeling. Kinetic simulations show that multiple remodelers can repeatedly occupy the same promoter region on a timescale of minutes, implicating an unending 'tug-of-war' that controls a temporally shifting window of accessibility for the transcription initiation machinery.

## Introduction

Eukaryotic chromatin is assembled in nucleosomes and higher order structures that compact the DNA for genome folding in the cell nucleus. Nucleosomes are actively organized at promoter and enhancer elements that are hypersensitive to nuclease digestion (*Almer and Hörz, 1986*; *Heintzman et al., 2007*; *Wu, 1980*). In the budding yeast *Saccharomyces cerevisiae*, gene promoters contain nucleosome-depleted regions (NDRs), approximately 150 base-pair stretches of DNA that are depleted of nucleosomes (*Yuan, 2005*). Non-canonical nucleosome conformations, often called 'fragile nucleosomes', and non-histone protein-DNA complexes, are also observed by limited MNase treatment and occupy a subset of NDRs (*Floer et al., 2010*; *Kubik et al., 2015*; *Prajapati et al., 2020*). NDRs are flanked by well-positioned +1 and −1 nucleosomes, with the +1 nucleosome overlapping the transcription start site (TSS) in yeast (*Albert et al., 2007*; *Yuan, 2005*). The +1 nucleosome also phases downstream nucleosome positions in regularly spaced locations which become progressively less well-positioned into the gene body (*Lai and Pugh, 2017*; *Mavrich et al., 2008*). This arrangement of nucleosomes is important for the accurate engagement of transcription regulators and the transcription pre-initiation complex [PIC], as well as the progression of the transcription machinery after initiation.

ATP-dependent chromatin remodelers are key *trans*-acting factors in establishing and maintaining nucleosome organization around genes (*Becker and Workman, 2013*; *Rando and Winston, 2012*; *Zhang et al., 2011*). As specialized members of the superfamily 2 (SF2) translocases, chromatin remodeling enzymes share a highly conserved ATPase motor that utilizes DNA translocation as the fundamental mechanism to restructure DNA-histone contacts within nucleosomes. In addition to the core ATPase domain, chromatin remodelers harbor additional functional domains and accessory sub-units, forming multiprotein complexes up to ~1 MDa in size that show substantial functional diversity. They are further classified into four sub-families based on sequence homology of the catalytic ATPase and possession of shared components, namely the SWI/SNF [Switch defective/sucrose non-fermenting], CHD [Chromodomain helicase DNA-binding], ISWI [Imitation switch], and INO80 [Inositol requiring 80] sub-families.

In vivo studies of remodelers in yeast revealed their distinct genome-wide specificities and functions in the multi-stage transcription process (*Yen et al., 2012*). In this context, remodelers can be distinguished based on their in vivo specificities for nucleosome targets genome-wide. The first group of remodelers, RSC, SWI/SNF, INO80, and ISW2, mainly act at gene promoter regions to define the +1 and −1 nucleosome positions. RSC and SWI/SNF mobilize the +1 and −1 nucleosomes away from the NDR relative to the TSS to promote proper engagement of transcription initiation machinery (*Ganguli et al., 2014*; *Klein-Brill et al., 2019*; *Kubik et al., 2018*). Specifically, RSC assists NDR formation for the majority of yeast genes, and the consequence of conditional RSC inactivation is a global loss of transcription (*Brahma and Henikoff, 2019*; *Ganguli et al., 2014*; *Kubik et al., 2018*; *Yen et al., 2012*). This has led to the concept of RSC (and SWI/SNF) as nucleosome 'pushers,' widening the NDR (*Kubik et al., 2019*). Antagonizing the pushing actions of RSC and SWI/SNF are INO80 and ISW2 (*Klein-Brill et al., 2019*; *Kubik et al., 2019*; *Shimada et al., 2008*; *Yen et al., 2012*). Both ISW2 and INO80 remodelers reposition the +1 and −1 nucleosomes towards the NDR in vivo, which is important for suppressing yeast cryptic transcription via noncanonical TSS usage (*Klein-Brill et al., 2019*; *Kubik et al., 2019*; *Whitehouse et al., 2007*). Furthermore, INO80 and ISW2 are implicated in DNA replication (*Au et al., 2011*; *Lee et al., 2015*; *Vincent et al., 2008*) and INO80 in DNA damage repair (*Morrison et al., 2004*; *van Attikum et al., 2004*).

The second group of remodelers, CHD1 and ISW1, act primarily in the gene body where they maintain proper nucleosome spacing and density relative to the +1 nucleosome. Their actions are coupled to transcription elongation by interacting with the elongating polymerase to maintain nucleosome density and thus suppress cryptic initiation within the gene body (*Cheung et al., 2008*; *Radman-Livaja et al., 2012*; *Smolle et al., 2012*). Remodelers with similar in vivo activities are functionally redundant as shown by stronger effects due to multiple deletions or depletions, compared to single deletion or depletion (*Kubik et al., 2019*; *Ocampo et al., 2016*). Furthermore, remodelers act competitively to fine-tune nucleosome positions around genes, leading to proper transcriptional regulation (*Kubik et al., 2019*; *Ocampo et al., 2016*; *Ocampo et al., 2019*; *Parnell et al., 2015*). These results further highlight the current perspective that nucleosomes located around genes are highly dynamic rather than static, and that the concerted actions of multiple remodelers result in the striking steady-state nucleosome organization observed by genome-wide mapping experiments. However, despite this knowledge, a gap still lies in our understanding of their real-time dynamics and timescales of remodeler interactions on their chromatin targets.

Here, we utilize single-molecule tracking (SMT) to directly observe and characterize the chromatin-binding kinetics of ATP-dependent chromatin remodelers in living cells (*Lionnet and Wu, 2021*). We investigated a comprehensive set of remodelers (RSC, SWI/SNF, CHD1, ISW1, ISW2, INO80) acting at gene promoter regions and gene bodies, allowing us to quantify and compare their in vivo dynamics. We show that remodelers have varying but substantial frequencies of chromatin binding, while exhibiting a common target search strategy of frequently engaging in highly transient (sub-second) chromatin interactions and stable residence times of only several seconds. We also discovered that the catalytic ATPase is responsible for enhancing their chromatin-associated diffusion and fast dissociation rates. By integrating the kinetic parameters measured for individual chromatin remodelers with values from genomic studies, we could simulate substantial temporal occupancies at yeast chromatin targets, leading to a tug-of-war model for the organization and dynamic positioning of the nucleosome landscape.

## Results

### Chromatin remodelers exist in chromatin bound and free populations

We tagged the catalytic subunits of six major chromatin remodeling complexes, RSC, SWI/SNF, CHD1, ISW1 (ISW1a, and ISW1b), ISW2, and INO80 at the C-terminus with the self-labeling HaloTag by engineering the endogenous loci and expressed the fusion proteins as the sole source under natural promoter control. The fusion proteins were localized in the nucleus and did not display detectable cleavage of the tag by SDS-PAGE (*Figure 1—figure supplement 1A,B*). Furthermore, no phenotypes were observed for all strains containing tagged constructs (*Figure 1—figure supplement 1C*). We then investigated their endogenous, real-time dynamics as representative subunits of chromatin remodeling complexes in asynchronous, log-phase cells (*Figure 1A*).

In order to quantify a broad range of kinetic behaviors displayed by remodelers, two imaging regimes were applied. 'Fast-tracking' acquires 10 ms frame-rate movies to directly measure a range of single-molecule diffusivities from 'slow' (chromatin-bound) to 'fast' (chromatin-free) and determine fractional representation (*Figure 1B*, *Figure 1—video 1*). However, high laser power and extensive photobleaching precludes measurement of chromatin residence times. 'Slow-tracking' with a longer 250 ms frame-rate and lower laser power motion-blurs fast diffusing molecules to selectively visualize the chromatin-bound state and report dwell times (*Figure 1C*). Combining the two imaging regimes provides a holistic and quantitative view of a range of diffusive behaviors and kinetic subpopulations.

We applied two independent methods for visualization and quantification of fast-tracking datasets. First, we determined the diffusion coefficient D for trajectories $\geq$ six frames (i.e. $\geq$ 60 ms) based on their mean squared displacements (MSD), and present frequency histograms based on the log(D) values of each trajectory. The histograms were initially fit to two Gaussian distributions, approximating slow and fast populations (*Figure 1—figure supplement 4*, *Figure 1—source data 1*). For more robust quantification, we applied Spot-On analytics, which uses kinetic modeling based on distribution of displacements for trajectories lasting $\geq$ three frames (*Hansen et al., 2018*; *Figure 1—figure supplement 2A,B*). Hereafter, we refer to diffusive values derived from Spot-On in the text. As previously reported for biological controls, H2B histone (Halo-H2B) and free HaloTag (Halo-NLS, nuclear localization signal), exhibit two distinct, well-separated diffusion states representing chromatin-bound and chromatin-free molecules (*Ranjan et al., 2020*). We found that the majority of H2B molecules (79.4 $\pm$ 1.9%) are slow-moving with average D of 0.026 $\mu m^2 s^{-1}$ (*Figure 1—figure supplement 2A*) consistent with incorporation into chromatin, whereas most of the chromatin-free Halo-NLS molecules show greatly increased diffusivity (D ~ 5 $\mu m^2 s^{-1}$) (*Ranjan et al., 2020*).

Compared to H2B, chromatin remodelers exhibit a slow $D_{bound}$ fraction (average 0.036 $\pm$ 0.007 to 0.067 $\pm$ 0.004 $\mu m^2 s^{-1}$) as would be expected for molecules associated with largely immobile chromatin (*Figure 1—figure supplement 2A*). However, as discussed later, the $D_{bound}$ values are ~2-fold higher than H2B. Furthermore, we also observed a separable chromatin-free fraction whose $D_{free}$ values (0.464 $\pm$ 0.043 to 1.014 $\pm$ 0.024 $\mu m^2 s^{-1}$) are ~10-fold higher, but distinctly lower than the $D_{free}$ for Halo-NLS, indicating that our imaging regime captures essentially the full range of potential diffusive behaviors for this family. In addition, the $D_{free}$ values show an inverse correlation with the estimated total molecular weights of chromatin remodeling complexes, consistent with expectations that the tagged catalytic subunits are associated within larger complexes (*Figure 1—figure supplement 1D*). We note that additional subpopulations, including variant or modified complexes, could be included within our two diffusive populations distinguishable by SMT.

We next assessed how the fractions of chromatin-bound and chromatin-free molecules vary among subgroups of chromatin remodeling enzymes. RSC and SWI/SNF mobilize +1 and −1 nucleosomes to increase promoter accessibility, while INO80 and ISW2 mobilize them to reduce accessibility (*Hartley and Madhani, 2009*; *Kubik et al., 2019*). We found that the majority of both RSC and SWI/SNF molecules are associated with chromatin (RSC: 66.0 $\pm$ 1.1%; SWI/SNF: 55.9 $\pm$ 1.3%) (*Figure 1D,E*). INO80 and ISW2 exhibit $F_{bound}$ values of 48.3 $\pm$ 0.2% and 34.8 $\pm$ 1.0%, respectively (*Figure 1F,G*). Overall, these NDR-acting remodelers display a broad range of chromatin-binding fractions (inclusive of stable and transient binding), with RSC showing the highest overall chromatin binding.

CHD1 and ISW1 act primarily on nucleosomes located in the gene body (*Kubik et al., 2019*; *Ocampo et al., 2016*). The two remodelers show comparable $F_{bound}$ values (CHD1: 47.8 $\pm$ 4.9%;

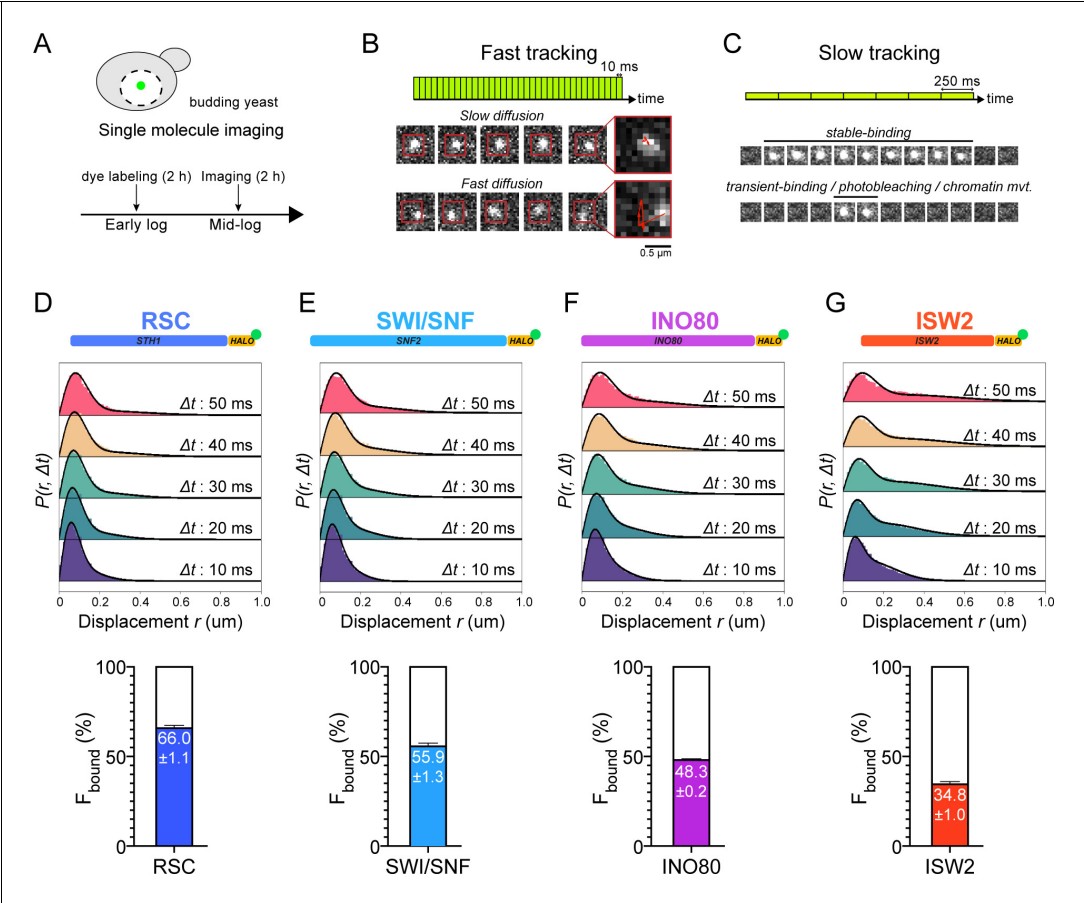

**Figure 1.** Chromatin-binding and chromatin-free fractions of RSC, SWI/SNF, INO80, and ISW2. (**A**) Experimental scheme. (**B**) Fast-tracking imaging regime uses short exposures (10 ms) at high laser power to distinguish slow (chromatin-bound) and fast (chromatin-free) diffusing populations. (**C**) Slow-tracking regime directly observes the dwell times of chromatin-bound molecules using 250 ms exposures at low laser power. (**D–G**) (Top) Raw displacement histograms over the first 5 time frames ($\Delta t$: 10, 20, 30, 40, 50 ms). A two-state kinetic model was used for fitting the CDF [black lines] in Spot-On. (Bottom) Spot-On kinetic modeling results based on displacement distribution histograms for Sth1-Halo (**D**), Snf2-Halo (**E**), Ino80-Halo (**F**), and Isw2-Halo (**G**). Solid colored bar with indicated value represents % chromatin-bound molecules; open bar represents % chromatin-free. Error bars are standard deviations from 2 [or 3 for ISW2] biological replicates.

The online version of this article includes the following video, source data, and figure supplement(s) for figure 1:

**Source data 1.** MSD-based kinetic analysis.

**Figure supplement 1.** Cell growth, integrity, and localization of HaloTagged remodeler subunits.

**Figure supplement 1—source data 1.** Original gel images for *Figure 1—figure supplement 1A*.

**Figure supplement 2.** Spot-On kinetic modeling analysis.

**Figure supplement 3.** Yeast culture during imaging and laser illumination do not have obvious effects on remodeler diffusion.

**Figure supplement 4.** $\log_{10}D$ histograms for six remodelers and Gaussian fitting.

**Figure 1—video 1.** Fast-tracking movie (10 ms exposure/frame) for Sth1-Halo strain.

https://elifesciences.org/articles/69387#fig1video1

ISW1a/b: 52.0 ± 1.1%) (*Figure 2A,B*). However, the catalytic subunit Isw1 is shared by two distinct chromatin remodeling complexes called ISW1a and ISW1b (*Vary, et al., 2003*), in addition to potentially un-complexed Isw1 catalytic subunit (*Tsukiyama et al., 1999*). The ISW1a complex localizes near the transcription start and end of genes, whereas the ISW1b complex occupies more mid-coding regions (*Morillon et al., 2003*; *Smolle et al., 2012*; *Yen et al., 2012*). Since Isw1 catalytic subunit dynamics represent a composite of the two remodeling complexes, we also tagged Ioc3 and Ioc4 accessory subunits unique to ISW1a and ISW1b complexes, respectively. The gene-body acting ISW1b (Ioc4-Halo) complex exhibits higher $F_{bound}$ compared to ISW1a (Ioc3-Halo) complex (ISW1b: 55.5 ± 2.6%; ISW1a: 39.6 ± 0.0%) (*Figure 2C,D*).

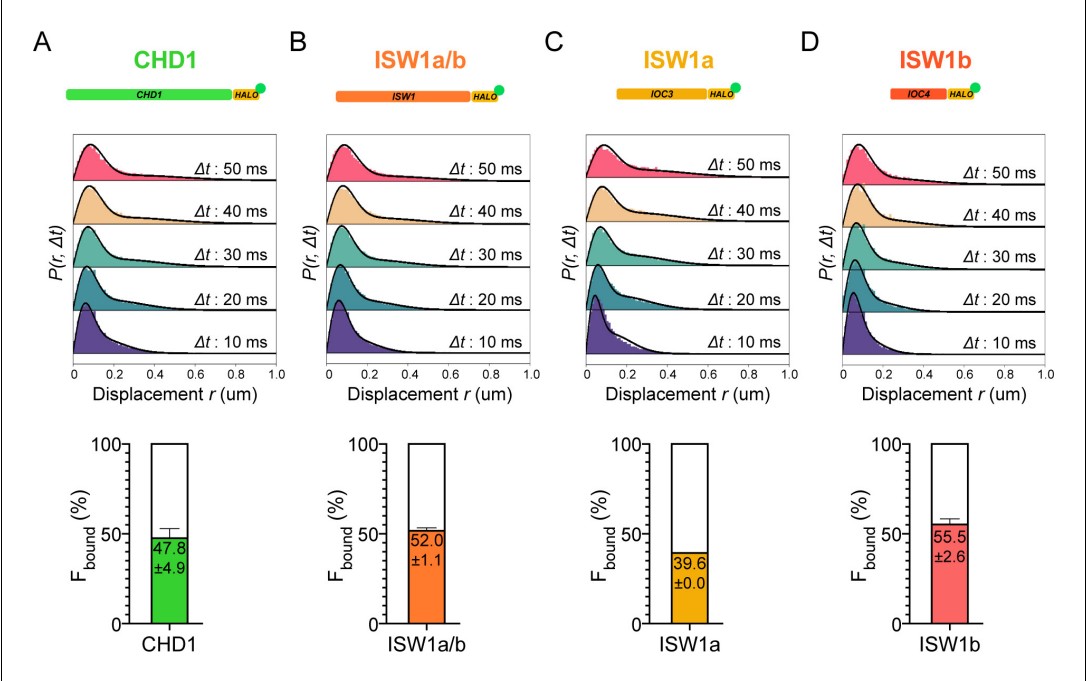

**Figure 2.** Chromatin-binding and chromatin-free populations of CHD1 and ISW1. (A–B) Spot-On analysis as described in *Figure 1* for the catalytic subunits Chd1-Halo (A) and Isw1-Halo (B). (C–D) Spot-On analysis of the accessory subunits of ISW1a and ISW1b complexes: Ioc3-Halo (C) and Ioc4-Halo (D).

## Chromatin remodelers frequently transition between bound and free states

Unlike the two well-separated Gaussian distributions for H2B histones and HaloTag protein (*Figure 3A*), the Log(D) histograms of all imaged chromatin remodelers display less distinct bound and free populations, with a noticeable fraction showing an intermediate range of diffusion coefficients (*Figure 1—figure supplement 4*). This population could either represent remodeler complexes transitioning between chromatin-bound and chromatin-free states, or chromatin-free molecules of intermediate diffusivity due to association with additional factors or confined inside a subnuclear compartment (*Hansen et al., 2020*; *Izeddin et al., 2014*; *McSwiggen et al., 2018*; *Strom et al., 2017*).

To distinguish between these possibilities, we analyzed single-particle trajectories using vbSPT, a variational Bayesian Hidden Markov Model (HMM) algorithm, which models state kinetics and assigns diffusive states to each displacement (*Persson et al., 2013*). We classified every displacement as either State 1 ('bound') or State 2 ('free') (*Figure 3—source data 1*), and sub-classified all trajectories as bound, free, or transitioning (*Figure 3B*). The median bound and free displacement lengths between transitioning and non-transitioning trajectories are highly similar or identical for each remodeler, validating the vbSPT state assignments and essentially excluding a dominant intermediate diffusive state (*Figure 3—figure supplement 1A–C*). Notably, the log D histograms of transitioning populations show enrichment for intermediate D values.

It is striking that the population of transitioning trajectories is more prominent for remodelers (from 20.9 ± 2.5% to 30.2 ± 1.5%) compared to free HaloTag (10.4 ± 0.7%) and H2B histone (7.1 ± 2.1%), (*Figure 3C–I*, *Figure 3—figure supplement 1D–E*). We observed comparable frequencies for remodeler dissociation (bound to free transition: 45.3 ± 1.3 to 50.24 ± 0.01%) and association (free to bound: 49.76 ± 0.01 to 54.7 ± 1.3%), indicating that there is little bias in the direction of state transitions (*Figure 3—figure supplement 1F*). Furthermore, the frequent detection of state transitions over short trajectory lifetimes suggests that the duration of each state is short-lived. We

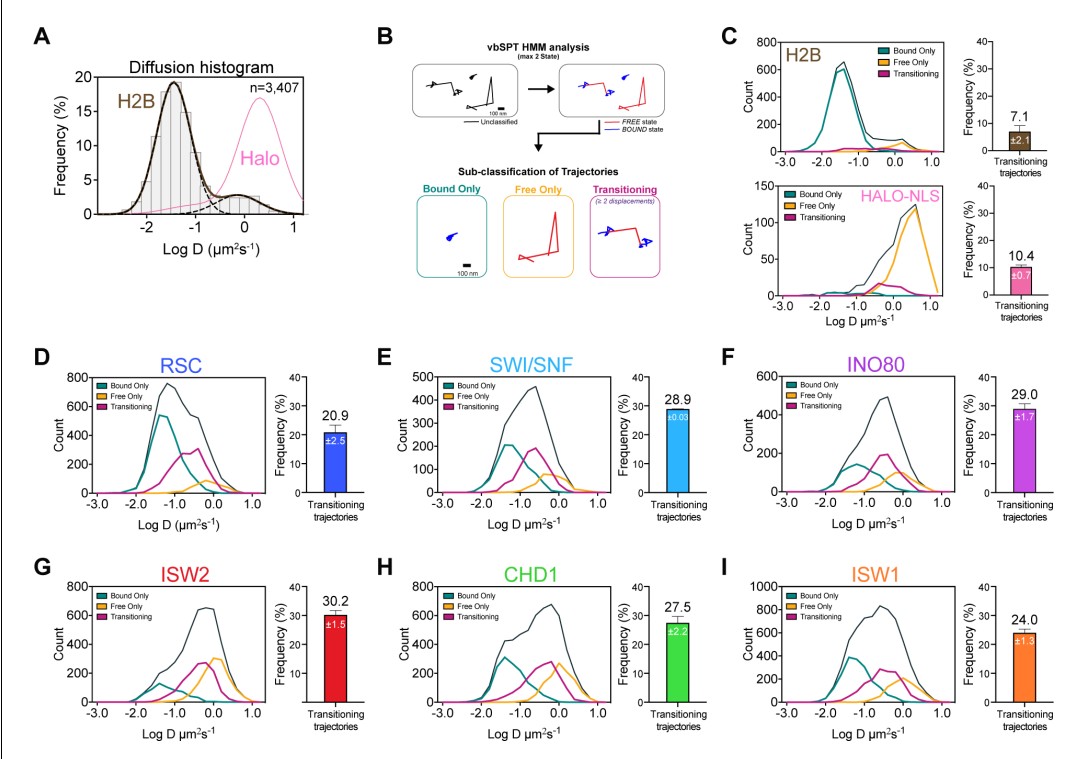

**Figure 3.** Remodelers undergo frequent transitions between bound and free states. (A) Halo-H2B (brown) and Halo-NLS (pink) molecules display well-separated peaks in their diffusion coefficient histograms. (B) An overview of displacement-based HMM classification (vbSPT) to identify transitioning trajectories. After classifying each displacement as either in bound or free state, each trajectory is sub-classified as 'bound only', 'free only', or 'transitioning'. (C–I) Left: Overlay of raw histograms of $\log_{10}$ diffusion coefficients for 'Bsound only' (turquoise), 'Free only' (yellow), 'Transitioning' (purple), and total trajectories (thin black). Right: Quantification (%) of transitioning trajectories in the diffusion coefficient histogram, where errors represent standard deviation between two [or three for ISW2] biological replicates. (C) Transitioning trajectories for Halo-H2B (top) and Halo-NLS (bottom). (D–I) Transitioning trajectories for remodelers: Sth1-Halo (D), Snf2-Halo (E), Ino80-Halo (F), and Isw2-Halo (G), Chd1-Halo (H), and Isw1-Halo (I). The online version of this article includes the following source data and figure supplement(s) for figure 3:

Source data 1. vbSPT analysis.

Figure supplement 1. Validation of two diffusive states classified by vbSPT, and quantification of transitioning frequencies.

concluded that transient but frequent chromatin interactions are characteristic of the six remodeling complexes.

## All remodelers have remarkably short in vivo residence times of 4–7 s

The chromatin-bound remodeler population measured by fast tracking consists of both transiently and stably bound molecules. We acquired long-exposure movies [250 ms/frame] under slow tracking (*Chen et al., 2014*) to generate survival curves revealing the apparent dissociation of chromatin-bound molecules as a function of time (*Figure 1C*). Particle dissociation can be due to molecules truly disengaging from chromatin, or to fluorophore photobleaching and chromatin movements out of focus, which can corrected using the survival curve of H2B histone as a standard (*Hansen et al., 2017*). The remodeler survival plots fit well to a double exponential decay model (*Figure 4—figure supplement 1A–F,H*), from which the average lifetimes ($\tau_{sb}$, $\tau_{tb}$) and fractions ($f_{sb}$, $f_{tb}$) of stable-binding and transient-binding species were extracted (*Figure 4*). All $\tau$ values presented in the text and figures are corrected based on H2B decay kinetics.

The stable-binding subpopulations ($f_{sb}$) of RSC (27 ± 2%) and SWI/SNF (24 ± 6%) display strikingly short lifetimes (RSC: 5.0 ± 0.7 s; SWI/SNF 4.4 ± 1.2 s) (*Figure 4A,B*), consistent with a previous measurement for the Rsc2 subunit of RSC (*Mehta et al., 2018*). Similarly, INO80 and ISW2 exhibit stable-binding fractions ($f_{sb}$ 20 ± 3% and 13 ± 3%, respectively) and similarly short residence times ($\tau_{sb}$

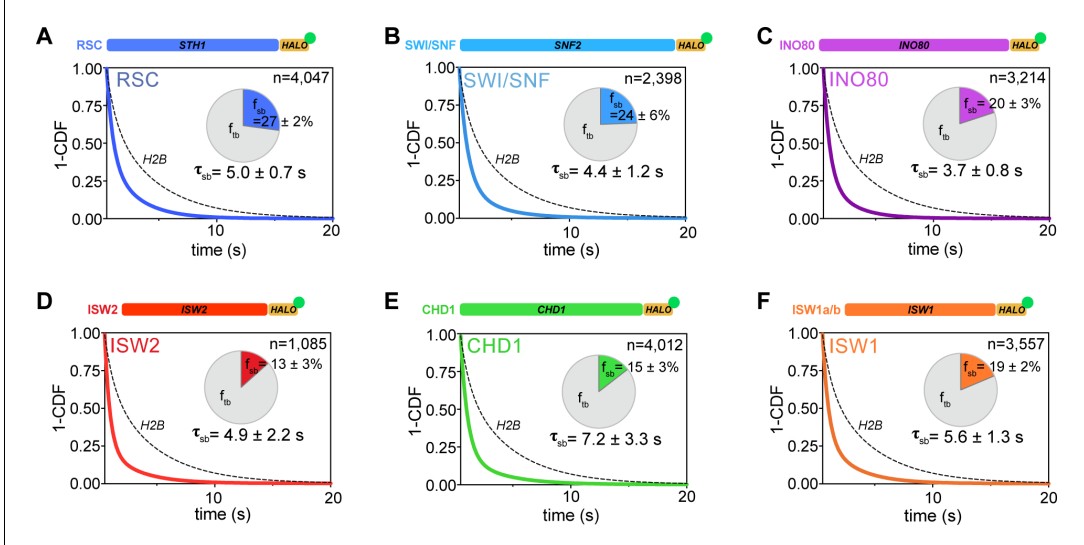

**Figure 4.** All remodelers have short-lived stable-binding residence times of 4–7 s. (A–F) Fitted double exponential decay curves from 1-CDF plots of observed dwell times from individual binding events (n) imaged by slow-tracking, for Sth1-Halo (A) Snf2-Halo (B), Ino80-Halo (C), and Isw2-Halo (D), Chd1-Halo (E), and Isw1-Halo (F). Solid colored and dashed black fitted curves for indicated remodelers and H2B, respectively. Pie charts show the percentage ($f_{sb}$) and average residence time ($\tau_{sb}$) of the stable binding population after photobleaching correction. Errors represent bootstrap resampling errors after resampling 100 times (sb: stable-binding; tb: transient-binding).

The online version of this article includes the following source data and figure supplement(s) for figure 4:

**Source data 1.** Kinetic parameters determined by slow-tracking.

**Figure supplement 1.** Survival plots [1-CDF] of dwell times showing 1- vs 2-component exponential decay fits.

---

3.7 ± 0.8 s and 4.9 ± 2.2 s, respectively) (**Figure 4C,D**). Hence, all NDR-acting remodelers bind stably for less than 5 s in live yeast, whereas transient-binding populations are more short-lived by almost an order of magnitude (**Figure 4—source data 1**).

For gene body-acting remodelers, CHD1 and ISW1 complexes exhibit stable-binding fractions ($f_{sb}$ 15 ± 3% and 19 ± 2%, respectively) and short dwell times ($\tau_{sb}$ 7.2 ± 3.3 s and 5.6 ± 1.3 s, respectively) (**Figure 4E,F**). Interestingly, ISW1b shows 2.5-fold higher residence times compared to ISW1a ($\tau_{sb}$ 5.9 ± 2.5 s and 2.2 ± 1.0 s, respectively) with comparable stable-binding fractions (**Figure 4—figure supplement 1G**). These remodelers also exhibit very short transient-binding residence times ($\tau_{tb}$< 0.65 s). Hence, the majority of chromatin binding events by remodelers is transient, and stable binding, on the order of several seconds, is notably short-lived.

## ATPase activity is coupled to fast dissociation rates

To examine whether the measured dissociation kinetics are intrinsic to chromatin remodeling complexes or functionally related to their ATP-dependent remodeling activities, we made strains harboring a point mutation in the ATPase domains of Isw2, Isw1, and Chd1; these mutations have previously been shown to abolish their ATPase activities (Isw2K215R, Isw1K227R, Chd1K407R, and Chd1D513N) (**Figure 5A**; *Fitzgerald et al., 2004*; *Gelbart et al., 2001*; *Hauk et al., 2010*; *Tsukiyama et al., 1999*). We then acquired slow-tracking movies to compare the dwell times of mutant to those of wild-type remodeling enzymes (**Figure 5B**).

We found that the stable-binding average residence time increased by ~2-fold (from 4.9 ± 2.2 to 9.7 ± 3.1 s) for the Isw2K215R mutant (**Figure 5C**). Similarly, we observed increased residence time (from 5.6 ± 1.3 to 9.1 ± 3.6 s) for the Isw1K227R (**Figure 5D**). The two ATPase-dead Chd1 mutants both showed increased stable-binding residence times (Chd1K407R from 7.2 ± 3.3 to ~52 s; Chd1D513N from 7.2 ± 3.3 to 16.1 ± 7.3 s) (**Figure 5E,F**). Interestingly, the tail of the Chd1K407R survival curve approaches that of H2B, which indicates its longevity, but precludes precise determination of dwell time (**Figure 5—figure supplement 1B**). All four mutants (**Figure 5—source data 1**) exhibit little to no changes in the transient-binding residence times compared to wildtype. In all, our

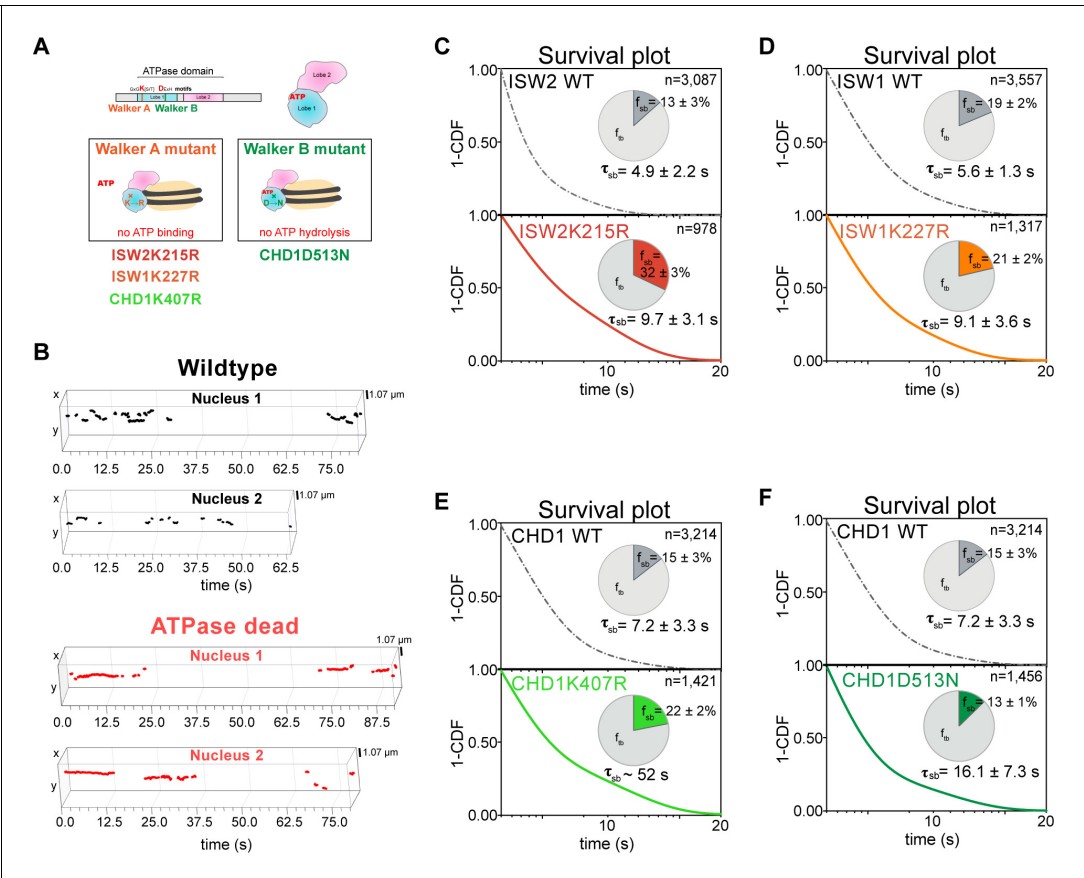

**Figure 5.** ATP hydrolysis is responsible for rapid chromatin dissociation. (**A**) Bar diagram and cartoons for remodelers mutated in the 'Walker A' and 'Walker B' motifs, respectively. (**B**) Representative 3D plots of trajectories imaged by slow-tracking for wildtype (Chd1-Halo, black) and ATPase-dead mutant (Chd1K407R-Halo, red). Each plot shows all trajectories (≥ three frames) from single nucleus where lines represent apparent durations of chromatin-binding events. (**C–F**) 1-CDF plot, pie chart, and residence times of wild-type (top) and ATPase-dead mutants (bottom) for Isw2 (**C**), Isw1 (**D**), and Chd1 (**E,F**).

The online version of this article includes the following source data and figure supplement(s) for figure 5:

**Source data 1.** Slow-tracking for ATPase-dead mutants.

**Figure supplement 1.** Expression levels and 1-CDF plots for wildtype and mutant ATPase-dead Isw2D312N.

**Figure supplement 1—source data 1.** Original gel images for *Figure 5—figure supplement 1A*.

results indicate that after chromatin association, the mutant ATPases exhibit slower dissociation rate (the reciprocal of residence time), consistent with previous genome-wide ChIP and biochemical studies (*Fitzgerald et al., 2004*; *Gelbart et al., 2001*).

## ATP binding enhances chromatin-bound mobility of remodelers

Chromatin imaged by several distinct methods in living cells displays heterogeneous mobility, which is dependent on its compaction state, subnuclear localization, and ATP-dependent processes (*Gasser, 2002*; *Gu et al., 2018*; *Marshall et al., 1997*; *Soutoglou and Misteli, 2007*). Remodelers may undergo 1D translocation on DNA (*Sirinakis et al., 2011*), and alter either local chromatin movement (*Basu et al., 2020*; *Neumann et al., 2012*) or higher order chromatin structure (*Lusser et al., 2005*; *Maier et al., 2008*) in an ATP-dependent fashion. We assessed the diffusive behavior of the chromatin-bound fraction of remodelers relative to the average dynamics of incorporated Halo-H2B histone. H2B not only reflects the motions of bulk chromatin but also provides an internal reference standard for nuclear movements and instrumental drift. From each trajectory classified as bound by vbSPT, the apparent D value and the $R_c$ [radius of confinement] were calculated to characterize its diffusivity and the confined domain encompassing the observed trajectory, respectively

(*Lerner et al., 2020*). Importantly, chromatin-bound remodelers exhibit ~2-fold higher mobility than H2B histone, as revealed by the average MSD plot and the distribution of individual D values of each trajectory under fast-tracking (*Figure 6A,B*). The mean $R_c$ values are also substantially higher for remodelers compared to the global mean measured for H2B (*Figure 6—figure supplement 1A*). This is further supported by the higher apparent D values to varying degrees [two- to fourfold] of stably-bound remodelers measured by slow-tracking (*Figure 6C*). Such greater mobility of chromatin-bound remodelers may be due to the combined effects of remodeler diffusion on chromatin and movement of the chromatin fiber caused by remodeling activity, or alternatively, may reflect the intrinsic dynamics of genomic loci being targeted.

To distinguish between these two alternatives, we measured the chromatin-associated mobility of the four aforementioned ATPase-dead mutants. Three mutants Isw2K215R, Isw1K227R, Chd1K407R harboring substitutions in the catalytic ATPase Walker A motif responsible for ATP binding display strikingly lower diffusivity as revealed by the average MSD plot of stably bound molecules, which

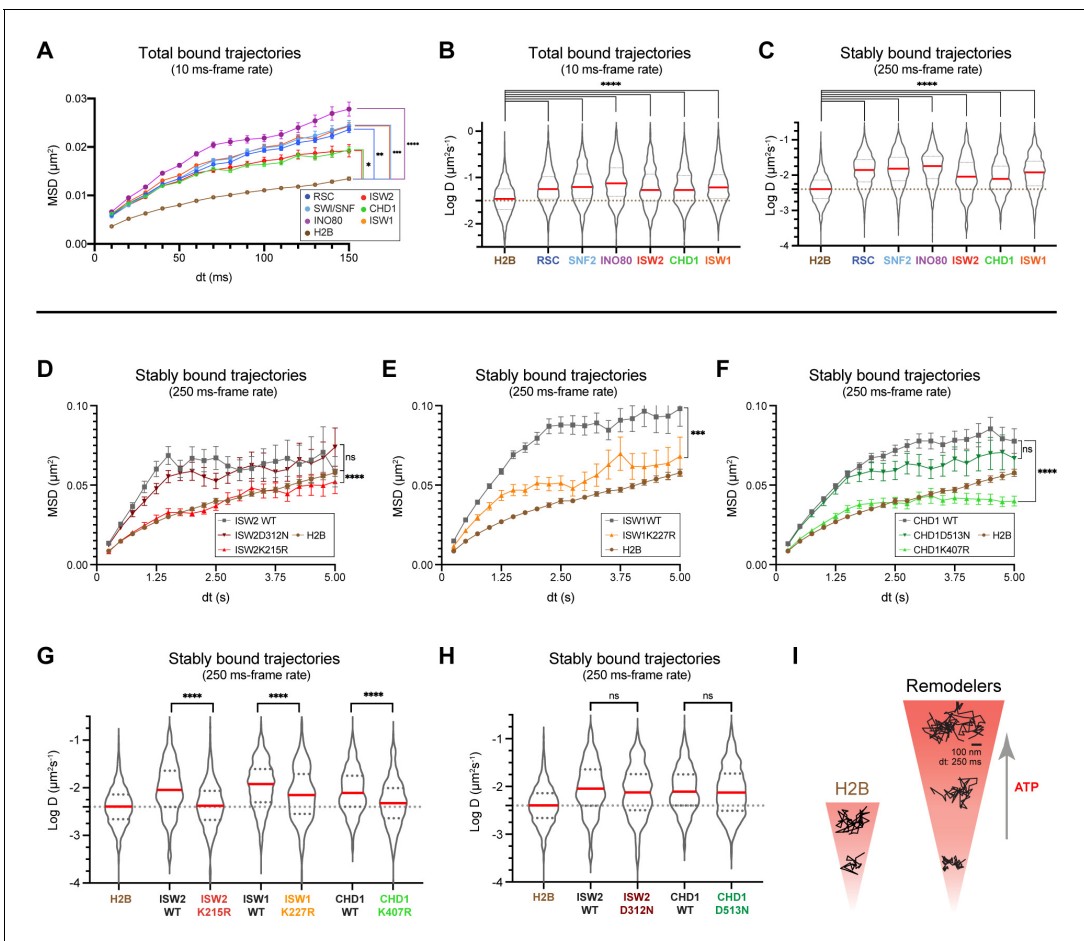

**Figure 6.** ATP utilization is responsible for enhanced mobility of chromatin-bound remodeler. (**A–B**) Average MSD plot (**A**) and violin plot (**B**), of individual D values for 'bound only' trajectories imaged by fast-tracking, shown for six remodelers and H2B histone. (**C**) Violin plot showing distribution of individual D values imaged by slow-tracking for six remodelers and H2B histone. For (**A–C**) each wildtype remodeler is compared to H2B by the ordinary one-way ANOVA test (****p<0.0001, ***p<0.001, **p<0.01, *p<0.05). (**D–H**) MSD plot (**D–F**) and violin plot (**G,H**) of individual D values for 'trajectories imaged by slow-tracking for wildtype, ATPase-dead mutant, and H2B. For violin plots, thick red and dotted gray lines represent the median and two quartiles, respectively. For **D–H**, mutants are compared to wildtype by the unpaired t test (****p<0.0001, ***p<0.001, ns: not significant). (**I**) Representative trajectories imaged by slow-tracking for H2B and remodelers. H2B displays low mobility, whereas remodelers display higher chromatin-associated diffusivity that is enhanced by ATP utilization.

The online version of this article includes the following source data and figure supplement(s) for figure 6:

**Source data 1.** Number of molecules (N), statistical tests, and source data for *Figure 6*.

**Figure supplement 1.** Chromatin-bound remodelers display higher radius of confinement ($R_c$) values than H2B.

approaches or substantially overlaps the global H2B curve (*Figure 6D–F*). This is supported by the violin plots of individual D values for stably bound trajectories (*Figure 6G*). Surprisingly, Chd1D513N bearing a substitution in the Walker B motif of Chd1 shows no substantial changes in the average MSD curve and apparent D values for stably bound trajectories compared to wildtype as measured by slow-tracking (*Figure 6F,H*). To rule out Chd1-specific effects, we also made a strain harboring the corresponding D-to-N substitution in ISW2 (Isw2D312N), and found no substantial changes in the average MSD curve and apparent D values compared to wildtype (*Figure 6D,H*). As expected for a mutation in the catalytic ATPase, the Isw2D312N mutant exhibits an approximately two-fold increase in its residence time compared to wildtype (*Figure 5—figure supplement 1C*). Because the Walker B motif is important for ATP hydrolysis (via coordinating $Mg^{2+}$ ion and a water molecule), but not for ATP binding (*Singleton et al., 2007*; *Walker et al., 1982*; *Figure 5A*), this result suggests that the ATP-bound state may be adequate to induce enhanced diffusion on chromatin as part of the mechanism of target search by remodeling enzymes.

## Promoter-enriched remodelers have robust chromatin occupancies

Chromatin remodelers are key regulators of the +1 nucleosome position genome-wide, whose accurate location is crucial for the PIC (pre-initiation complex) formation and TSS fidelity (*Lai and Pugh, 2017*; *Zhang et al., 2011*). RSC and SWI/SNF mobilize the +1 nucleosome away from the NDR, opposed by INO80 and ISW2 activities, which slide the +1 nucleosome towards the NDR. As a quantitative indicator of nucleosome engagement, we determined the occupancies of the four remodelers, that is the percent average occupancy at a chromatin target by each remodeler over a given time period. To calculate temporal occupancy, we utilized the measured overall chromatin-binding fraction [$F_{sb}$] and the temporal parameters for stable [$\tau_{sb}$, $f_{sb}$] and transient [$\tau_{tb}$, $f_{tb}$] chromatin-binding (*Figure 7A*). Here, we assume that stable binding, which is almost an order of magnitude longer than transient binding, represents binding at 'specific' target sites within promoter regions including

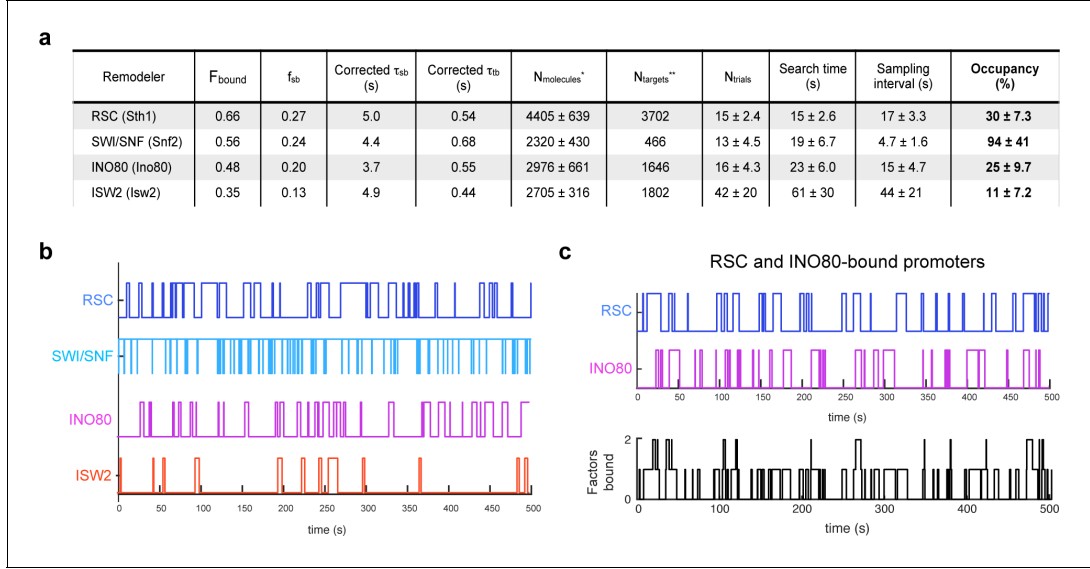

**Figure 7.** Remodelers show substantial temporal occupancies at chromatin targets. (A) Key parameters measured in this study and acquired from the literature (*Ho et al., 2018*; *Kubik et al., 2019*) are used to calculate occupancy levels for gene promoter-acting remodelers. (B) Time trace simulations of temporal occupancy for individual remodelers at a target promoter region based on average $\tau_{sb}$ and sampling interval. Top and bottom bars represent occupied (on) and vacant (off) states, respectively, and vertical lines depict transitions between the two states. (C) Time trace simulations of occupancy at a RSC- and INO80-bound promoter region based on average $\tau_{sb}$ and sampling interval. Individual time trace simulations are shown above, and the cumulative simulated occupancy plot (black) shows either one or both remodelers bound in the time course of 500 s.

The online version of this article includes the following figure supplement(s) for figure 7:

**Figure supplement 1.** Time trace simulations of temporal occupancies at promoters bound by multiple remodelers, and analysis of CHD1 DNA-binding mutant.

−1, +one [+1] nucleosomes and the intervening NDR, while transient binding represents non-specific chromatin interactions (*Ball et al., 2016*).

The fraction of stable-binding depends on both the number of molecules per nucleus ($N_{molecules}$) and number of the target sites in the genome ($N_{targets}$) (*Chen et al., 2014*). We rely on published values for $N_{molecules}$ compiled from a meta-analysis of 21 proteomic studies for the best available quantitative estimate of remodeler abundance in *S. cerevisiae* (*Ho et al., 2018*), with the assumption that our sole source HaloTag fusions under natural promoter control are similar in abundance to the untagged proteins. As regards the number of chromatin targets ($N_{targets}$), many studies have investigated the genome-wide specificities of chromatin remodelers using multiple approaches including ChIP-Seq (*Cutler et al., 2018*; *Floer et al., 2010*; *Shimada et al., 2008*; *Spain et al., 2014*), MNase-ChIP (*Yen et al., 2012*), ChIP-exo (*Rossi et al., 2021*), Native-ChIP-Seq (*Ramachandran et al., 2015*; *Zentner et al., 2013*), CUT and RUN (*Brahma and Henikoff, 2019*), and ChEC-seq (*Kubik et al., 2019*). For this paper, we utilized the $N_{targets}$ values reported by Kubik et al., who investigated the binding sites for four yeast remodelers of interest in this study, and explicitly quantified the number of mRNA gene promoters enriched for each remodeler. Assuming that the +1 or −1 nucleosomes and the NDR together represent the main interaction substrates at promoter regions, this value of $N_{targets}$ can be considered as a lower bound estimate. Accordingly, RSC binding is the most widespread ($N_{targets}$ = 3702), SWI/SNF binds only a small subset ($N_{targets}$ = 466), and INO80 ($N_{targets}$ = 1646) and ISW2 ($N_{targets}$ = 1802) each bind to approximately a third of all yeast promoters (*Kubik et al., 2019*). Because INO80 and ISW2 are also targeted to sites of DNA replication, as well as tDNA and ribosomal DNA loci (*Cutler et al., 2018*; *Gelbart, 2005*; *Shimada et al., 2008*), the total $N_{targets}$ for each remodeler is likely to be larger than the values we utilized.

To calculate occupancy values, we used the $\tau_{search}$ (search time), the time it takes for a molecule to go from one stable target site to the next [i.e. time bound non-specifically plus time in free diffusion], the SI (sampling interval) [i.e. the time between initial binding of one molecule and binding of the second molecule], and the estimated values for $N_{molecules}$ and $N_{targets}$ per cell (*Figure 7A*; see Materials and methods). RSC shows substantial occupancy (30 ± 7.3%) at stably bound chromatin targets, despite its short residence time ($\tau_{sb}$) of 5.0 ± 0.7 s. Thus, rather than individual RSC molecules residing for long periods of time, the high occupancy rate can be attributed to the short $\tau_{search}$ (15 ± 2.6 s) and comparable SI (17 ± 3.3 s) values coupled to high $N_{molecules}$ (>4000) (*Figure 7A*). SWI/SNF exhibits the highest occupancy (94 ± 41%) among the four remodelers, consistent with the highest raw ChEC signals reported for Swi3 (*Kubik et al., 2019*). Strikingly, our estimate indicates that SWI/SNF also maintains its occupancy at target sites by coupling short residence time (4.4 ± 1.2 s) with short SI (4.7 ± 1.6 s) (*Figure 7A*).

We next assessed the occupancy values for INO80 and ISW2, which oppose the actions of RSC and SWI/SNF. Comparable to RSC, INO80 displays substantial occupancy (25 ± 9.7%) at its targets while ISW2 displays a lower occupancy (11 ± 7.2%). INO80 exhibits short $\tau_{search}$ (23 ± 6.0 s) and SI (15 ± 4.7 s) values, while ISW2 has a relatively longer $\tau_{search}$ (61 ± 30 s) and SI (44 ± 21 s) values (*Figure 7A*). Average time trace simulations of stably bound occupancies for each of the four remodelers over several hundred seconds (*Figure 7B*) show that at promoter regions targeted by multiple remodelers such as genes in 'cluster IV' enriched for RSC and INO80 (*Kubik et al., 2019*), the occupancy by any one remodeler is strikingly high, and more than one remodeler can simultaneously engage a promoter repeatedly over several minutes (*Figure 7C*; see also *Figure 7—figure supplement 1A,B*). (Note that our calculated promoter occupancy values for INO80 and ISW2 represent an upper limit and are subject to revision depending on the extent of remodeler binding to sites of DNA replication or repair).

## Discussion

Imaging chromatin remodeler diffusion by the fast-tracking mode in yeast shows that they bind to chromatin at substantial frequencies [$F_{bound}$: 35–66%], and with a notable population [21–30%] displaying intermediate D values resulting from transitions between bound and free states (*Figures 1–3*). This suggests remodelers frequently undergo highly short-lived chromatin interactions and is consistent with our slow-tracking measurements of transient-binding frequency ($f_{tb}$: 73–87%) (*Figure 4*), and with FRAP and FCS measurements of over-expressed mammalian ISWI (*Erdel et al., 2010*). The high frequency of transient interactions, and direct evidence for transitioning trajectories is also

consistent with the model of 1D-3D facilitated diffusion, a proposed mechanism to increase the target search efficiency of nuclear proteins (*von Hippel and Berg, 1989*), By slow-tracking, two chromatin-associated populations, 'stable-binding' and 'transient-binding', were observed for all six remodelers. Previous SMT studies on the mammalian Sox2 and yeast Ace1 transcription factors showed that stable-binding subpopulation represent interactions with cognate target sequences (*Chen et al., 2014*; *Mehta et al., 2018*). Comparable to reported values for the Rsc2 subunit of RSC and the yeast transcription factors, Ace1 and Gal4 (*Donovan et al., 2019*; *Mehta et al., 2018*), all imaged remodelers show stable and transient residence times of 4–7 s and 0.4–0.7 s, respectively. Furthermore, the effect of mutating the DNA-binding domain of CHD1 monomer (*Ryan et al., 2011*; *Tran et al., 2000*) results in a threefold reduction in the $\tau_{sb}$ value (from $7.2 \pm 3.3$ to $2.4 \pm 0.7$ s) (*Figure 7—figure supplement 1C,D*). Unlike sequence-specific transcription factors, a complete loss of stable-binding would not be expected for remodeling complexes, whose recruitment relies on multiple interactions with gene-specific transcription factors, histone modification recognition domains, and interaction with components of the transcription machinery (*Becker and Workman, 2013*). Indeed, we speculate that the multiplicity of interaction motifs has a central role in the unusual diffusive behaviors shown by chromatin remodelers.

Importantly, the fast dissociation rates of remodelers are facilitated by ATP hydrolysis. Five tested ATPase-dead mutants (for ISW2, ISW1, CHD1) show twofold or greater increase in their stable-binding residence times (*Figure 5C–F*), highlighting a new role of for ATP-utilization in coupling nucleosome remodeling to rapid enzyme dissociation from chromatin. This also suggests that their mean residence times can reflect timescales for the diverse reactions performed by remodeling enzymes on chromatin in vivo. Assuming that the +1 or −1 nucleosomes and the NDR are the main targets for promoter-acting RSC, SWI/SNF, INO80 and ISW2, their 4–7 s stable residence time would include time for diffusion on the NDR as well as time expended for nucleosome remodeling. Biochemical studies have shown that remodelers undertake small translocation steps with remodeling rates of a few bp/sec (*Blosser et al., 2009*; *Deindl et al., 2013*; *Harada et al., 2016*; *Qiu et al., 2017*; *Sabantsev et al., 2019*). For example, with an enzymatic rate of 2 bp/s for ISWI (*Blosser et al., 2009*), an ISWI stable-binding event would allow octamer sliding by roughly 12 bp, which is within range of in vivo nucleosome position changes after conditional inactivation of RSC, SWI/SNF, INO80, and ISW2 (*Ganguli et al., 2014*; *Kubik et al., 2019*).

Under our imaging conditions, the chromatin-bound populations of the six remodelers exhibit higher mobility than H2B measured for bulk incorporated histones (*Figure 6A–C*). We further showed that this enhanced mobility is dependent on the ATPase domain. Mutations in the ISW1, ISW2, and CHD1 Walker A motif implicated in nucleotide binding (*Singleton et al., 2007*) substantially decreases in vivo mobility. Further analysis of mutations in the CHD1 and ISW2 Walker B motif, implicated in ATP hydrolysis [but not ATP binding] (*Singleton et al., 2007*), displays a milder decrease of its chromatin mobility, suggesting that nucleotide binding is largely sufficient for promoting diffusion for the two remodelers. Previous observations for other DNA-binding ATPase enzymes have noted ATP binding-dependent, hydrolysis-independent 1D diffusion or sliding on DNA (*Cho et al., 2012*; *Mazur et al., 2006*; *Tóth et al., 2015*), suggesting that this mode of diffusion to enhance target search may be shared among remodelers. In addition, RSC and *Drosophila* ISWI remodelers undergo ATP hydrolysis-dependent translocation on ssDNA and dsDNA in vitro (*Saha et al., 2005*; *Whitehouse et al., 2003*), with processivities of 20–70 bp/translocation event (*Fischer et al., 2007*; *Saha et al., 2005*; *Sirinakis et al., 2011*; *Whitehouse et al., 2003*). Finally, the absence of any change in ISW1 chromatin-bound mobility upon treatment with a general transcription inhibitor thiolutin rules out transcription per se as a source of enhanced remodeler diffusion (*Figure 6—figure supplement 1B*). In all, our results suggest that chromatin remodelers use the catalytic ATPase not only for nucleosome remodeling but also to enhance target search kinetics by promoting 1D diffusion on chromatin and rapid detachment after reaction.

Yeast promoter regions can be classified into different groups enriched either for no remodeler or a combination of RSC, SWI/SNF, INO80, and ISW2, with about half of promoters genome-wide harboring at least two distinct remodelers that harbor nucleosome pushing and pulling activities relative to the NDR (*Kubik et al., 2019*). At promoter regions where opposing remodelers bind, we expect a consecutive 'tug-of-war' between the pushing and pulling activities, in which successive engagements would ultimately result in fine-tuning the steady-state nucleosome position, with the final outcome dependent on remodeler occupancy and nucleosome remodeling activity. Based on

occupancy estimates, two remodelers may be found to simultaneously engage promoter chromatin (*Figure 7C*, *Figure 7—figure supplement 1A,B*), but steric considerations likely preclude two remodelers binding to the same nucleosome or the same face of a nucleosome. Alternatively, at promoter regions where none or only one remodeler binds, other mechanisms are likely to have more substantial roles in nucleosome positioning. These include the sequence-dependent bendability of promoter DNA as well as the binding of general regulatory factors (GRFs), such as Reb1, Abf1, and Rap1, acting as barriers to nucleosome mobility (*Struhl and Segal, 2013*).

## A temporal model for nucleosome remodeling at NDRs

By integrating our live-cell SMT measurements with available genome-wide localization and protein expression data, we estimate temporal occupancies ranging from 11 ± 7.2 to 94 ± 41% for RSC, SWI/SNF, INO80, and ISW2 at target promoter regions including the NDR and flanking nucleosomes. Our findings of highly dynamic and frequent remodeler-nucleosome interactions are consistent with recent genomics studies showing substantial changes in nucleosome positions upon rapid, conditional inactivation of remodelers in yeast and mammalian systems (*Iurlaro et al., 2021*; *Klein-Brill et al., 2019*; *Kubik et al., 2019*; *Schick et al., 2021*). Accordingly, we envision a nucleosome remodeling cycle at promoters in which remodeler combinations undergo frequent association, ATP-dependent mobilization and dissociation from chromatin to dynamically fine-tune −1 and +1 nucleosome positions (*Figure 8*).

We anticipate stochastic recruitment of RSC, SWI/SNF, INO80 and ISW2 to their target promoter regions. RSC recognizes general promoter characteristics, such as the long DNA stretch of the NDR (*Wagner et al., 2020*), histone acetylation marks potentially read by eight bromodomains in four RSC subunits (*Josling et al., 2012*), and the Rsc3 DNA-binding sequence motif found in several hundred promoters (*Badis et al., 2008*). These recruitment mechanisms likely account for RSC enrichment at the majority of yeast promoters [3702/5040] (*Kubik et al., 2019*). For SWI/SNF recruitment to a minority of yeast promoters [466/5040 promoters], extensive studies have shown interactions with gene-specific transcription factors (*Cosma et al., 1999*; *Neely et al., 1999*; *Peterson and Workman, 2000*; *Yudkovsky et al., 1999*). ISW2 also interacts with transcription factors such as Ume6 (*Goldmark et al., 2000*), but overall, less is known about the recruitment mechanisms for INO80 and ISW2, presenting opportunities for future studies.

Upon binding within the accessible NDR, RSC or SWI/SNF undergoes 1-D diffusion in an ATP-dependent manner, manifesting higher chromatin-associated mobility. On engagement with either flanking nucleosome substrate [+1 nucleosome shown], RSC or SWI/SNF uses the energy of ATP hydrolysis to reposition the nucleosome away from NDR, enlarging NDR length. Importantly, this remodeling activity also facilitates RSC or SWI/SNF dissociation. Subsequent stochastic recruitment of INO80 or ISW2, ATP-dependent 1-D diffusion, and nucleosome engagement remodels the nucleosome to move in the opposing direction and narrow the NDR, coupled with remodeler dissociation. Cycles of sequential or simultaneous binding and activity by the four remodelers with their similar dwell times (4–5 s) and varying sampling intervals (5–44 s) provides a dynamic temporal window of accessibility for promoter chromatin.

In a related study (*Nguyen et al., 2021*), the average promoter occupancy of the yeast PIC that forms upstream and overlapping the +1 nucleosome was found to be in the range of 10%, that is on the same order of magnitude but lower than three of four remodelers examined. Similar to chromatin remodelers, a full PIC lasts only several seconds before dissociation from chromatin, but the average promoter is vacant for ~100 s before PIC reformation. Thus, we suggest that there may be robust and dynamic competition between PIC components and mobilized NDR-flanking nucleosomes with chromatin exposure of key promoter elements such as the TATA box occurring for only a limited time window allowing proper assembly of downstream PIC components. This temporally positioned +one [+1] nucleosome would enable Pol II to scan and start transcription at the proper, canonical TSS. In this way, the dynamic interactions of remodeling enzymes with their promoter targets provides a temporal, chromatin accessibility-based regulatory mechanism for eukaryotic transcription.

Taken together, our SMT study elucidates the dynamic behaviors of this family of nuclear proteins and offers insights into additional kinetic functions for the remodeling ATPase and the timescales that govern nucleosome repositioning in relation to transcription events. It is notable that an independent study from the Verrijzer laboratory (Tilly et al. 2021) found that the *Drosophila* Brahma

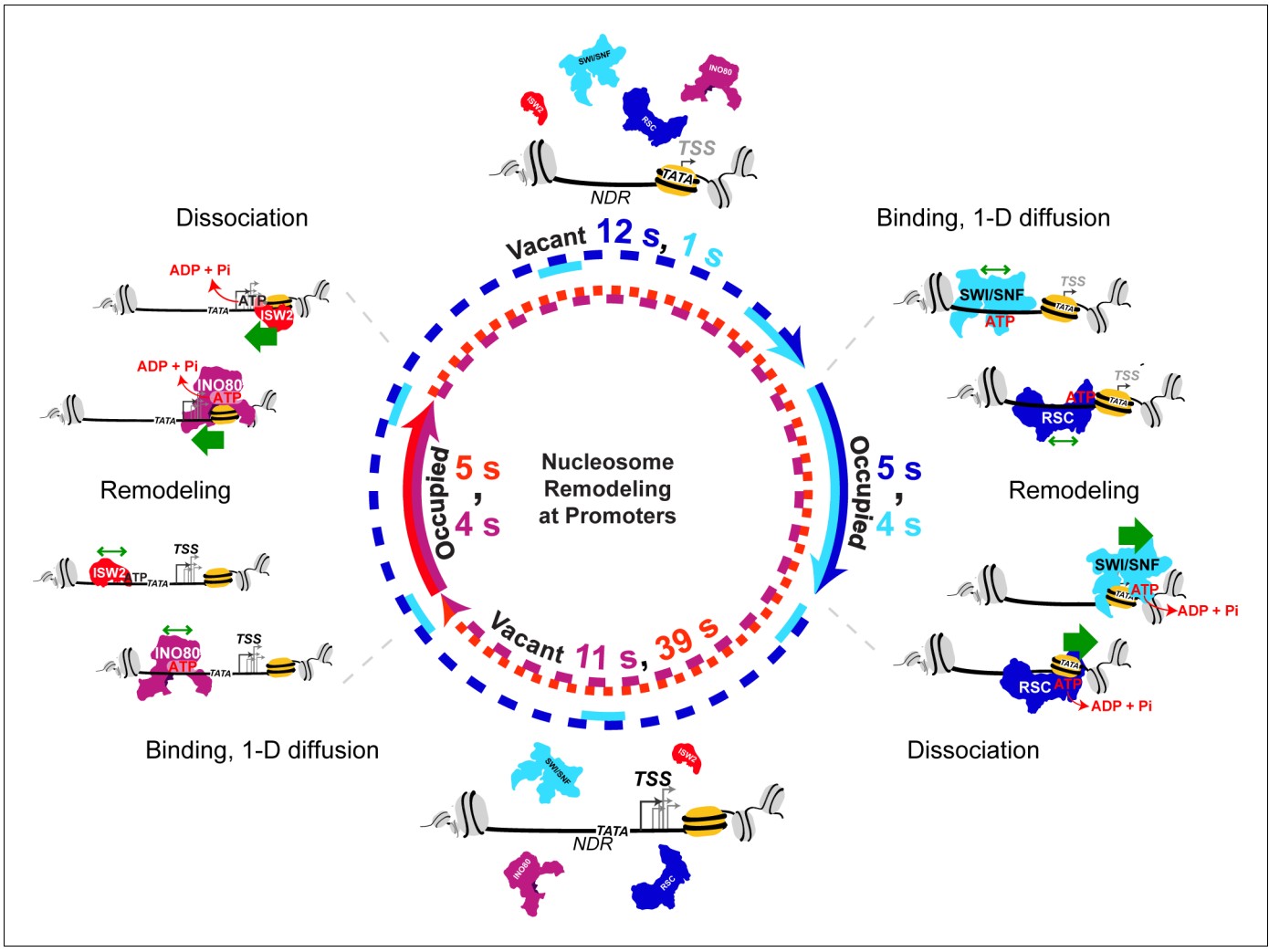

**Figure 8.** Nucleosome remodeling at promoters. Model for nucleosome remodeling cycle at a gene promoter region targeted by RSC, SWI/SNF, INO80, and ISW2. The promoter region transitions between remodeler-occupied [solid arrow] and remodeler-vacant [dashed arrow] states, and their mean durations are indicated. After association with the NDR, remodelers undergo 1-D diffusion on chromatin facilitated by ATP binding, resulting in higher chromatin-associated mobility. Upon engaging a nucleosome substrate [e.g. the +1 nucleosome], RSC or SWI/SNF uses the energy of ATP hydrolysis to 'push' the nucleosome away from the NDR and INO80 or ISW2 to 'pull' the nucleosome into the NDR. ATP hydrolysis facilitates remodeler dissociation, and the promoter region becomes vacant for other factor interactions. The order of remodeler visitation is arbitrary, and simultaneous co-occupancy within the NDR can occur (see text for details).

(BRM) remodeler also displays dependence on ATP utilization by the catalytic ATPase for dynamic mobility and remodeler dissociation in live *Drosophila* cells. For INO80 and ISW2, it would be of interest to distinguish their diffusive behaviors specific to DNA replication in cells undergoing S phase, and to DNA repair in the presence of DNA damaging agents. Outstanding questions also include determining the kinetic parameters for other chromatin regulators such as histone acetyltransferases, methyltransferases, histone de-modification enzymes, and histone chaperones, to gain a comprehensive view of the overall competition for engagement of promoter-proximal nucleosomes, their effects on nucleosome positioning and the dynamics of transcription complexes. Our findings in live cells provide a temporal framework for further testing of proposed models and should facilitate development of in vitro single-molecule assays that allow direct observation of physical and functional interactions between transcription regulators, chromatin, and the transcription machinery.

# Materials and methods

**Key resources table**

| Reagent type (species) or resource | Designation | Source or reference | Identifiers | Additional information |
|---|---|---|---|---|
| Strain, strain background (*S. cerevisiae*) | Full list of yeast strains is presented in *Supplementary file 1*. | | | |
| Recombinant DNA reagent | pBS-SK-Halo-NatMX (plasmid) | *Ranjan et al., 2020* | | |
| Recombinant DNA reagent | pUG72 (plasmid) | Euroscarf, Germany | pUG72 (P30117 )-Euroscarf | |
| Sequence-based reagent | Full list of oligonucleotides is presented in *Supplementary file 2*. | | | |
| Chemical compound, drug | JF552-HaloTag ligand | *Zheng et al., 2019* | N/A | |
| Chemical compound, drug | JF646-HaloTag ligand | *Grimm et al., 2015* | N/A | |
| Chemical compound, drug | #1.5 Micro Coverglass – 25 mm Diameter | Electron Microscopy Sciences | 72225–01 | |
| Software, algorithm | ImageJ (1.52 p) | ImageJ | RRID:SCR_003070 https://imagej.net/ | |
| Software, algorithm | Diatrack 3.05 | *Vallotton and Olivier, 2013* | http://www.dia track.org/index.html | |
| Software, algorithm | GraphPad Prism version 8.4.2 | GraphPad Software, Inc. | RRID:SCR_002798 http://www.graphpad.com | |
| Software, algorithm | Sojourner package | Carl Wu lab | https://rdrr.io/github /sheng-liu/sojourner/ | |
| Software, algorithm | Spot-On | *Hansen et al., 2018* | https://spoton.berkeley.edu/ | |
| Software, algorithm | vbSPT | *Persson et al., 2013* | http://vbspt.sourceforge.net/ | |
| Software, algorithm | Radius of confinement calculation | *Lerner et al., 2020* | https://data.mendeley .com/datasets/wctzwpp9h2/2 | |
| Software, algorithm | Custom Matlab script | This paper, Mendeley Data | http://dx.doi.org/ 10.17632/ydwcx9yhpp.2 | |

## Yeast strains

All *Saccharomyces cerevisiae* strains used in this study are isogenic derivatives of W303 strain carrying *pdr5△* for efficient JF ligand labeling, and are listed in *Supplementary file 1*. HaloTag was fused to the C-terminus of the protein of interest using standard methods for yeast transformation, using pBS-SK-Halo-NatMX plasmid (*Ranjan et al., 2020*). Point mutations were introduced by either the traditional 'pop-in pop-out' (*Rothstein, 1991*) or the '50:50' method (*Horecka and Davis, 2014*), using pUG72 plasmid (P30117, Euroscarf, Germany), and list of primers are provided in *Supplementary file 2*.

## Yeast growth assays

The cell growth of strains carrying HaloTag fusion constructs were compared to their derived parental genotype. The strains grown to saturation in YPAD (Yeast Extract-Peptone-Dextrose + 40 mg/L Ade-SO$_4$) were serially diluted (5-fold dilutions) on YPAD plates. Plates were imaged after 48 or 72 hr growing at three different temperatures (25℃, 30℃, and 38℃).

## Cell lysate preparation to check integrity of HaloTag fusion proteins

Yeast cultures growing at early log phase (OD600 0.2) were treated with JF646 dye at a saturating dye concentration of 20 nM JF646 (*Grimm et al., 2015*) was used instead of JF552 for better dye labeling (*Ranjan et al., 2020*). Yeast protein extract was prepared using the NaOH method (*Amberg et al., 2006*). Total protein concentration was measured using the Bradford Assay, and 45 ng of total protein was loaded per well in SDS-PAGE. Gels were imaged on Tecan five scanner, with Cy5 excitation. After imaging, gels were stained with Coomassie dye for loading control.

## Yeast culture preparation for single molecule imaging

Yeast cultures growing in Synthetic Complete Medium (0.79 g/L Complete Supplement Mixture [CSM] Powder, Sunrise Science Products, Cat. No. 1001–010; 6.7 g/L Yeast nitrogen base without amino acids, BD Difco, Cat. No. DF0919-15-3; 2% (w/v) dextrose; 40 mg/L Adenine hemisulfate) were treated with dyes at early log phase (D600 0.2–0.3) for 2 hr. For fast-tracking, saturating dye concentrations ranging from 10 to 20 nM JF552 (*Zheng et al., 2019*) were used depending on factor abundance. For slow-tracking, we used 5–7.5 nM JF552. In some instances, we also added JF646 (~5 nM) to visualize nuclear fluorescence without JF552 excitation and to partially reduce JF552 label-ling. Cells were harvested around mid-log phase by brief centrifugation (3500 rpm for 2 min), washed at least three times, and finally resuspended in CSM medium. Resuspended cells were loaded on Concanavalin A-treated coverslip (#1.5 Micro Coverglass −25 mm Diameter, Electron Microscopy Sciences, Cat. No. 72225–01) assembled on imaging cell chamber (Invitrogen, Cat. No. A7816), where coverslips were flamed prior to the treatment in order to reduce single-to-noise back-ground. After immobilization and additional washing, ~1.5 mL of fresh medium was added to the chamber in which the cells are bathed continually for the duration of the imaging session (~2 hr). Comparison of results from the first and second halves of an imaging session shows no substantial differences (*Figure 1—figure supplement 3A*).

## Live cell, single molecule imaging using wide-field microscopy

### Microscope setup

All yeast imagings were performed using a custom-built Zeiss widefield microscope (Axio Observer Z1) with a 150X glycerin immersion objective (NA 1.35) as previously described (*Ranjan et al., 2020*). Data was acquired with EM-CCD (Hamamatsu C9100-13) camera with FF01-750/SP and NF03-405/488/561/635E quad-notch filters for a final x-y pixel size of 107 nm. All imagings were per-formed with a single excitation channel. For JF552 dye excitation, 555 nm laser (Crystalaser) at (TTL pulsed) with 561 beam-splitter and 612/69 nm filter was used. For JF646 dye excitation, a 639 nm laser with 648 beamsplitter and 676/29 nm filter was used. Microscope manipulations (i.e. Z-focus, X/Y translation, filter cube switch) was performed by Zen software (Zeiss, Germany) and camera and data acquisition was controlled by HCImage software (Hamamatsu Photonics, Japan).

## Data acquisition

After yeast immobilization, the asynchronous cells representing all phases of the cell cycle were imaged for around 2 hr at room temperature.

### Fast tracking

Movies with 10 ms exposure/frame were recorded with continuous 555 nm laser irradiation at ~1 kW/cm$^2$. A field of view of 128 x 128 pixels was used to capture 4–6 yeast nuclei. Single-molecule imaging was performed using dSTORM (direct stochastic optical reconstruction microscopy) (*Heilemann et al., 2008*; *Rust et al., 2006*). Each movie begins with global nuclear fluorescence on laser excitation of essentially all JF552-labelled molecules before photoconversion to a dark, non-fluorescent state. The duration of the initial excitation pulse which pushes fluorescent JF552 to the dark state is ~10 s, depending on the density of labeled Halo-proteins, which is a function of protein expression and dye labeling concentration adjusted for the specific protein. Once in the dark state, JF552 spontaneously and stochastically converts back to the fluorescent state (*Grimm et al., 2015*; *Zheng et al., 2019*).This conversion is sufficiently slow such that mostly one JF552 molecule per yeast nucleus is fluorescent at any given time during data acquisition, allowing for an unambiguous spatio-temporal record of single JF552-labeled proteins. To ensure that we can harvest a 'substack'

of at least 5000 frames (10 ms/frame) (See Single molecule image analysis), we routinely record movies of ~1.5 min. About 40 movies were acquired per imaging session, and two [or three for ISW2] biological replicates were obtained for each sample.

Under fast-tracking conditions, cells displayed no detectable cellular damage and underwent normal bud growth and cell division when examined every 30 min up to 3 hr (*Nguyen et al., 2021*). Furthermore, we did not observe any substantial difference between the first and second half of each movie (*Figure 1—figure supplement 3B*).

## Slow tracking

Movies with 250 ms exposure/frame were acquired using continuous 555 nm laser irradiation at 0.05 kW/cm$^2$ (5% of fast-tracking power) for sufficient signal-to-noise while minimizing photobleaching. A focal plane of 256 x 256 pixels was used to capture 15–20 yeast nuclei. In the beginning of each movie, the 639 nm excitation channel was briefly used to fine-tune the focus, and then immediately switched to 555 nm excitation to start data acquisition. Upon 555 nm laser illumination, we record ~5 min movies starting with fluorescent JF552 pushed to the dark state for > 30 s, followed by single-molecule imaging of at least 750 frames at 250 ms/frame. Fifteen to twenty movies were taken per imaging session, with two or three biological replicates for wildtype and two to four biological replicates for mutant strains.

## Single molecule image analysis

For each raw movie, we first manually selected a 'substack' where ~one single molecule per nucleus per frame was observed in order to minimize tracking errors resulting from connecting different molecules as one trajectory. Substack lengths of 5000 frames (50 s) and 750 frames (3.125 min) were selected for fast and slow tracking movies, respectively, using ImageJ (1.52 p) custom-written script. The substacks were then applied to the Diatrack software (ver. 3.05, http://www.diatrack.org/index.html) to localize the centroid of PSF (point spread function) by Gaussian fitting over the fluorescence intensity to sub-pixel resolution (*Thompson et al., 2002*; *Yildiz et al., 2003*) and track single particles (*Vallotton and Olivier, 2013*). For localization, the following parameters were applied: Remove dim: 75–85, Remove blurred: 0.1, Activate High Precision mode: ON (HWHM=one pixel). For tracking, we used max jump of '6' (642 nm) and '3' pixels (321 nm) for fast and slow tracking datasets, respectively. Furthermore, we masked the nuclear regions based on the maximum intensity Z-projection of the selected substacks to filter out trajectories found outside of the nucleus in the subsequent analysis steps. The Diatrack output file containing information about the x, y coordinate and frame number were then applied for further downstream analysis.

## Fast tracking

MSD-based diffusion coefficient histograms: All 'masked' trajectories with at least five displacements were analyzed, using the lab custom-written R package, Sojourner (https://rdrr.io/github/sheng-liu/sojourner/; *Liu et al., 2020*). Briefly, for each trajectory, MSD plot for time lags from 2 to 5 $\triangle$t ($\triangle$t = 10 ms) were generated, then fit to linear regression (filtering out $R^2$ < 0.8 plots). From the slope, the diffusion coefficient was calculated as (where d is the number of dimensions, or 2):

$$D = \frac{1}{2d} \times \frac{MSD(dt)}{dt}$$

Spot-On (*Hansen et al., 2018*): All 'masked' trajectories with at least two displacements were analyzed. The following parameters were applied for Jump Length Distribution: Bin width (μm): 0.01, Number of time-points: 6, Jumps to consider: 4, Use entire trajectories No, Max jump (μm): 2. Additionally, the following parameters were applied for 2-state Model Fitting: $D_{bound}$ (μm$^2$/s): 0.0005–0.1, $D_{free}$ (μm$^2$/s): 0.15–25, $F_{bound}$: 0–1, Localization error (μm): Fit from data (0.01–0.1), dZ (μm): 0.6, Use Z correction, Model Fit: CDF, Iterations: 3.

vbSPT (variational Bayesian) HMM (*Persson et al., 2013*): All 'masked' trajectories with at least two displacements were analyzed. The following parameters were used to run vbSPT-1.1.3 to classify each displacement into two states, 'Bound' or 'Free' (*Hansen et al., 2020*; *Hansen, 2019*, https://gitlab.com/anders.sejr.hansen/anisotropy): timestep = 0.01; dim = 2; trjLmin = 2; runs = 3;

maxHidden = 2; stateEstimate = 1; bootstrapNum=10; fullBootstrap = 0; init_D = [0.001, 16]; init_tD = [2, 20]*timestep; and default prior choices according to *Persson et al., 2013*.

Then each trajectory was sub-classified as 'Bound only' if all displacements are classified as bound state; 'Free only' if all displacements are classified as free state; and 'Transitioning' if the trajectory contains both bound and free displacements with at least two consecutive displacements in each state. To validate that the transitioning trajectories consist of bound and free states, we calculated and compared the displacement length between 'bound only' and bound segments of transitioning trajectories, and between 'free only' and free segments of transitioning trajectories. Finally, the sub-classified trajectories were used to regenerate the diffusion coefficient histograms.

Radius of confinement: All 'masked' trajectories with at least four displacements were analyzed, as described previously (*Lerner et al., 2020*). To determine the radius of confinement exhibited by chromatin-bound molecules, we analyzed trajectories classified as 'bound only' by vbSPT (as described above). Since many confined trajectories with low D do not pass the $R^2 \geq 0.8$ filter, we used all trajectories whose MSD plots passed the more lenient $R^2 \geq 0.1$ filtering. The MSD plot was then fit to the circular confined diffusion model:

$$MSD_{circle} = R^2 \cdot \left(1 - e^{\frac{-4 \cdot D \cdot t_{lag}}{R^2}}\right)$$

where R is the radius of confinement, D is the short-term diffusion coefficient. Specifically, the first 10 time points of the MSD plot were used to fit to the model, and trajectories with squared norm of residual (RSS) higher than $10^{-5}$ and Rc higher than 300 nm were discarded.

## Slow tracking

Residence times: Using Sojourner package, the apparent lifetimes (temporal length of trajectories) were determined for all 'masked' trajectories lasting at least three frames. To account for blinking or mislocalizations, we allowed for gaps up to two frames between two localizations and linked them as one trajectory if they were less than three pixels apart. 1-CDF curves were generated and fit to a double exponential decay model:

$$P(t) = f_{sb}e^{-k_{sb}t} + f_{tb}e^{-k_{tb}t}$$

where $k_{sb}$ and $k_{tb}$ correspond to dissociation rates for stable- and transient-binding events, respectively, and $1 = f_{sb} + f_{tb}$ for the two components.

The apparent $k_{sb}$ and $k_{tb}$ values are affected by technical and imaging limitations such as photobleaching and chromatin movements. To correct for this bias, we used apparent dissociation rates of H2B imaged under same conditions as described previously (*Hansen et al., 2017*). The corrected residence times for stable- ($\tau_{sb}$) and transient binding ($\tau_{tb}$) were calculated as follows:

$$\tau_{sb} = \frac{1}{k_{sb} - k_{sb,H2B}}$$

$$\tau_{tb} = \frac{1}{k_{tb} - k_{sb,H2B}}$$

Apparent diffusion coefficient values for stably bound trajectories: All 'masked' trajectories lasting at least five frames (not allowing for gaps) were analyzed, using Sojourner package. For each trajectory, MSD plot for time lags from 2 to 5 $\triangle t$ ($\triangle t$ = 10 ms) were generated, then fit to linear regression (filtering out $R^2 < 0.8$ plots). From the slope, the diffusion coefficient was calculated as (where d is the number of dimensions, or 2):

$$D = \frac{1}{2d} \times \frac{MSD(dt)}{dt}$$

## Occupancy calculation

To calculate temporal occupancy, we integrated approaches from previous studies (*Chen et al., 2014*; *Loffreda et al., 2017*; *Tatavosian et al., 2018*).

Search time ($\tau_{search}$) is the average time it takes for a molecule to go from one specific site to its next specific site. The two specific binding events (lasting for $\tau_{sb}$) are interspersed by a number of trials ($N_{trials}$) binding to non-specific sites (lasting for $\tau_{tb}$). $\tau_{free}$ is the average free time between two binding events. Assuming equal probability of binding to all specific and non-specific sites, the search time is calculated as follows:

$$\tau_{search} = N_{trials} \times \tau_{tb} + (N_{trials} + 1) \times \tau_{free}$$

$N_{trials}$ depends on the ratio of number of non-specific ($N_{ns}$) to specific sites ($N_s$), or $r_s$:

$$N_{trials} = \frac{N_s + N_{ns}}{N_s} = 1 + r_s$$

Here, $r_s$ can be determined based on two assumed scenarios for bound molecules observed during slow tracking (as described in *Nguyen et al., 2021*). First, $f_{sb}$ determined by slow tracking depends on the time a molecule spends bound to specific sites compared to nonspecific sites:

$$f_{sb} = \frac{N_s \times \tau_{sb}}{N_s \times \tau_{sb} + N_{ns} \times \tau_{tb}} = \frac{\tau_{sb}}{\tau_{sb} + r_{s,1} \times \tau_{tb}}$$

Thus $r_s$ is equal to:

$$r_{s,1} = \frac{\tau_{sb}}{\tau_{tb}} \times \left(\frac{1}{f_{sb}} - 1\right)$$

In the second scenario, $f_{sb}$ depends on the probability that a free molecule binds to a specific site over all sites:

$$f_{sb} = \frac{N_s}{N_s + N_{ns}} = \frac{1}{1 + r_{s,2}}$$

In this case $r_s$ is:

$$r_{s,2} = \frac{1}{f_{sb}} - 1$$

We take the average value calculated from the 2 proposed scenarios to finally determine $r_s$:

$$r_s = \frac{1}{2}\left(\frac{1}{f_{sb}} - 1\right)\left(\frac{\tau_{sb}}{\tau_{tb}} + 1\right)$$

In fast tracking, $F_{bound}$ is percentage or fraction of the time a molecule spends bound to chromatin either specifically or non-specifically:

$$F_{bound} = \frac{N_{trials} \times \tau_{tb} + \tau_{sb}}{N_{trials} \times \tau_{tb} + \tau_{sb} + (N_{trials} + 1) \times \tau_{free}}$$

Thus $\tau_{free}$ is (in terms of $r_s$):

$$\tau_{free} = \frac{\frac{(1+r_s) \times \tau_{tb} + \tau_{sb}}{F_{bound}} - (1 + r_s) \times \tau_{tb} - \tau_{sb}}{2 + r_s}$$

Using the values derived for $r_s$ and $\tau_{free}$, we then calculated the search time as shown above.

Sampling interval (SI) is the time interval between two specific binding events at a given site as described previously (*Chen et al., 2014*):

$$Sampling\ Interval(SI) = \frac{(\tau_{search} + \tau_{sb}) \times N_{targets}}{N_{molecules}}$$

We used $N_{targets}$ values presented by *Kubik et al., 2019*. $N_{molecules}$ was determined as the median and standard error values (*Ho et al., 2018*), and their standard error was used for error propagation.

Finally, occupancy is the temporal probability that a given specific site is occupied by the protein of interest:

$$Occupancy = \frac{\tau_{sb}}{SI}$$

## Target occupancy simulation

Remodeler occupancy at a target promoter region was simulated as described previously (*Nguyen et al., 2021*). Briefly, experimentally determined $\tau_{sb}$ and estimated sampling interval (SI) values were used to simulate sequential promoter-occupied and vacant states over the time trace (500 s). The duration for each occupied and vacant state was randomly chosen from exponential distributions of the average $\tau_{sb}$ and (SI- $\tau_{sb}$) values, respectively. For promoter regions targeted by multiple remodelers, each remodeler was independently subject to the occupancy simulation, and the number of any single or multiple remodeler(s) co-occupying each timepoint was calculated throughout the time trace.

## Acknowledgements

We thank Vu Q Nguyen, Anand Ranjan, and Gaku Mizuguchi for experimental guidance at the initial stages of this project, Sun Jay Yoo, Yick Hin Ling and Taibo Li for computational assistance, Pascal Vallotton for support with Diatrack software, Jonathan Lerner, Ken Zaret, and Melike Lakadamyali for assistance with the Two-parameter single-molecule analysis, Slawomir Kubik and David Shore for advice on analysis of genomic data on remodeling enzymes, Toshio Tsukiyama and Brad Cairns for yeast strains, Anders Hansen and Greg Bowman for discussions, Wu lab members for helpful comments, and Peter Verrijzer for sharing a pre-print of his study. This study was supported by funds from a Korean Foundation for Advanced Studies Fellowship (JMK), a Johns Hopkins Bloomberg Distinguished Professorship (CW) and National Institute of Health grant GM132290-01 (CW).

## Additional information

### Competing interests

Qinsi Zheng, Luke D Lavis: LDL and QZ are listed as inventors on patents and patent applications whose value might be affected by publication. US Patent 9,933,417 and Patent Application 2021/0085805 describing azetidine-containing fluorophores and variant compositions (with inventors QZ, LDL, and TL) are assigned to HHMI. Timothee Lionnet: TL holds intellectual property rights related to Janelia Fluor dyes used in this publication. US Patent 9,933,417 and Patent Application 2021/0085805 describing azetidine-containing fluorophores and variant compositions (with inventors QZ, LDL, and TL) are assigned to HHMI. The other authors declare that no competing interests exist.

### Funding

| Funder | Grant reference number | Author |
|---|---|---|
| National Institutes of Health | GM132290 | Carl Wu |
| National Institutes of Health | GM127538 | Timothee Lionnet |
| Korea Foundation for Advanced Studies | | Jee Min Kim |
| Johns Hopkins Bloomberg [Johns Hopkins University] | | Carl Wu |

The funders had no role in study design, data collection and interpretation, or the decision to submit the work for publication.

### Author contributions

Jee Min Kim, Conceptualization, Software, Formal analysis, Supervision, Investigation, Writing - original draft, Writing - review and editing; Pat Visanpattanasin, Vivian Jou, Formal analysis, Investigation; Sheng Liu, Xiaona Tang, Timothee Lionnet, Software; Qinsi Zheng, Luke D Lavis, Resources; Kai Yu

Li, Jonathan Snedeker, Investigation; Carl Wu, Conceptualization, Supervision, Funding acquisition, Writing - original draft, Writing - review and editing

**Author ORCIDs**
Pat Visanpattanasin (ORCID) https://orcid.org/0000-0002-9506-8360
Carl Wu (ORCID) https://orcid.org/0000-0001-6933-5763

**Decision letter and Author response**
Decision letter https://doi.org/10.7554/eLife.69387.sa1
Author response https://doi.org/10.7554/eLife.69387.sa2

## Additional files

**Supplementary files**
• Supplementary file 1. List of yeast strains used in this study.
• Supplementary file 2. List of oligonucleotides used in this study.
• Transparent reporting form

**Data availability**
All custom scripts and imaging data files have been deposited in Mendeley Data and are publicly available via: https://data.mendeley.com/datasets/ydwcx9yhpp/2 (https://doi.org/10.17632/ydwcx9yhpp.2).

The following dataset was generated:

| Author(s) | Year | Dataset title | Dataset URL | Database and Identifier |
| --- | --- | --- | --- | --- |
| Kim JM | 2021 | Kim et al. 2021 | https://doi.org/10.17632/ydwcx9yhpp.2 | Mendeley Data, 10.17632/ydwcx9yhpp.2 |

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
