## [Decision Letter]

**Acceptance summary:**

In this study, Kim and co-workers track the dynamics of a large set of different ATP-dependent chromatin remodelers in living cells by utilizing state-of-the-art single-molecule imaging. They report that the remodelers exhibit very high turnover rates at target loci/nucleosomes, find evidence for cooperativity among the remodelers, and reveal the role of ATP hydrolysis in those interactions. These observations allow the authors to put forward a model for tug-of-war activities that modulate the accessibility of promoter regions for transcriptional activity. This manuscript brings important new information to the remodeler and chromatin dynamics field.

**Decision letter after peer review:**

Thank you for submitting your article "Single-molecule imaging of chromatin remodelers reveals role of ATPase in promoting fast kinetics of target search and dissociation from chromatin" for consideration by *eLife*. Your article has been reviewed by 3 peer reviewers, and the evaluation has been overseen by Sebastian Deindl as the Reviewing Editor and Kevin Struhl as the Senior Editor. The following individuals involved in review of your submission have agreed to reveal their identity: Xiaojun Ren (Reviewer #1); Bradley R Cairns (Reviewer #2).

Essential revisions:

1) The impact of the paper would be increased if the model presented in Figure 8 could be extended to include all six remodelers studied, thereby highlighting their kinetics similarities and the resulting tug-of-war at promoters.

2) The authors should comment on the perceived discrepancy in the relative abundance of the remodelers. The authors should consider adding the proposed quantification via Western Blot if feasible.

3) The authors should comment on several important technical points raised by the reviewers, in particular in terms of the imaging conditions (single remodelers rather than clusters, potential effect of exposure to laser light, media flow during data acquisition) and data analysis (decomposition of histograms in Figures 1 and 2. The authors should also comment on the cell cycle phase and explicitly describe if/how it was taken into account.

Appropriate textual changes to the manuscript would improve clarity on these points.

*Reviewer #1 (Recommendations for the authors):*

1) It appears to this reviewer that the Log D and jump distance histograms in Figures 1 and 2 should be decomposed into three components: chromatin-bound, intermediate diffusive, and free diffusive, since the authors have identified an intermediate diffusive population that can transition between bound and free states, which are shown in Figure 3.

2) The authors use the number of molecules of remodelers from published results (Ho, 2018) to calculate target-search kinetics. It appears to this reviewer that the published data does not match the data in Figure S1A. For instance, Figure 7A shows that INO80 is more abundance than ISW2; however, Figure S1A shows the opposite results. Given that these remodelers have been endogenously tagged, it would be possible to quantify the number of molecules of remodelers through western blotting by comparing to known concentrations of HaloTag.

3) P27L603: It should be 10 ms exposure time.

4) P27L605: Please explain why each video lasts 1.5 min.

5) P27L607: It should be 250 ms exposure time.

6) P27L612: Please explain again why each video lasts 7-9 min.

7) P28L638: Please check the equation.

8) P32L716: Please check the reference.

*Reviewer #2 (Recommendations for the authors):*

The model presented in figure 8 restricts itself to two of the remodelers (RSC and INO80) addressed, and does not reflect the full scope of the results obtained in this work. It would be helpful if changes could be made to the model figure in order to embrace all six remodelers studied, highlighting their kinetics similarities, and the resulting tug-of-war at promoters. Notably, it would be helpful to consolidate and clarify the ATPase state, as ATP and ADP+Pi are written concomitantly at all stages of each remodeler presented. Moreover, the model would benefit from a more clear depiction of ATP binding in the enhancement of nucleosome browsing, and of ATP hydrolysis in the enhancement of dissociation.

*Reviewer #3 (Recommendations for the authors):*

The paper is a technical tour-de-force, but I have a number of technical questions that need clarification. These arise partially from my ignorance, but also from the fact that there seem to be details missing in the methods section. Given that the paper should be read by a maximal number of chromatin biologists, I think it would greatly enhance the impact of the paper to include explanations concerning the following points, either in the methods section or in the text itself.

1. How do they know that they are actually imaging SINGLE chromatin remodelers in every case, and not clusters of a given remodeler, for example, bound at multiple nucleosomes at a promoter (or cluster of promoters) ? I understand that they use a pulsed laser to activate the Halo tag fluorochrome, but it seems to me that they need to quantify the number of molecules each signal might represent. Hansen and Darzacq have worked out means to do this. The authors could also estimate the number of total molecules of each remodeler this way, which would be useful to know, especially to see if this influences occupancy rates.

2. The pixel size (over 100 nm) is quite large – certainly large enough to contain multiple protein complexes. How large are the signals they monitor ? Did the try to reduce pixel size (it is possible with the LSM 710 and 100x objective)?

3. The authors use an imaging scheme worked out by Hansen et al. for "rapid" and "slower" movements. In one they image at 10ms intervals for 1.5 min, but for the slower one they image at 250ms intervals for 7-9 minutes. This latter scheme implies a lot of exposure to laser light. Did they check that there was no impact on cell division rate after the slower, longer imaging protocol ? It is a massive dose, and the time needed for cell division should be monitored under these conditions.

4. The authors say that they image for 2 h after the immobilization of cells on a cover slide. Are the cells continually bathed in medium during imaging ? this should be stated, if so. During 2 hours without glucose yeast rapidly deplete their ATP levels and this strongly influences chromatin mobility and remodeler activity (as shown 20 years ago, but also in this paper by walker motif mutation). It leads one to wonder if the slower moving remodeler tracks were taken at the end of the 2h period and the faster ones at the early time points ? Or did they keep a flow of fresh media onto the cells ? This is a very important technical point that should be explained in the methods.

5. For the longer (but not shorter) tracking and subsequent calculation of diffusion coefficients the authors should be conscious that nuclear movement (or cellular and instrumental drift) can alter the track, if not normalized out. I guess the fact that they subtract H2B movement does normalize for cellular or instrumental drift, but perhaps they should state that. Other means exist, e.g. to track the center of the nucleus interpolated from a ring of nuclear pores or background nuclear fluorescence – and subtract that movement from the movement of the SPT. Some discussion of removal of such background drift should be mentioned in methods, or if this has already been ruled out elsewhere, the reference should be cited.

6. Finally, the authors assume that the nucleosome remodelers are almost exclusively involved in transcription, and based on this they make the tug-of-war argument that INO80 and ISWI2 are counteracting RSC and Swi/SNF action. However, the two remodelers that have slightly different diffusion histograms (INO80 and ISWI2) are both implicated in DNA replication (and also in DNA repair), as well as transcription. The Tsukiyama lab has clearly demonstrated their roles in replication, and various authors provided convincing evidence that INO80 is important for paused fork restart and other repair events. DNA repair may be too rare an event (compared to transcription) to bias the movement observed, but replication could definitely bias the mobility/diffusion or Spot-on profiles, if a fraction of the cells imaged were in S phase.

This raises a number of questions:

a) Were the cells imaged exclusively in G1 phase ? That is, were only cells without buds monitored ? This should be stated if so.

b) If not, did they monitor INO80 or ISWi2 diffusion specifically in S phase cells and did this lead to the same or a different profile of chromatin-bound/-free fractions and diffusion co-efficients?

c) The diffusion histograms are fit to two Gaussian curves, but they would also be compatible with the superposition of multiple populations. Given that INO80 (and probably ISWI2) respond to phosphorylation, and may be regulated by cell cycle phase – it might be that there are more than simply a "bound" and an "unbound" population. This should – at the very least be discussed.

I think it should be mentioned that these specific two remodelers are involved in replication – and what that might imply – unless all cells imaged were in G1 phase. If they were all G1 phase cells, then it should have been mentioned (sorry if I missed it).

d) There could also be multiple variant complexes, containing additional subunits or being post-translationally modified, that might account for the complexity of Diffusion coefficients. This should at least be mentioned somewhere in the discussion.

With clarity on these issues, I think this paper definitely brings important new information to the remodeler and chromatin dynamics field.

---

## [Author Response]

Essential revisions:1) The impact of the paper would be increased if the model presented in Figure 8 could be extended to include all six remodelers studied, thereby highlighting their kinetics similarities and the resulting tug-of-war at promoters.

We agree that the impact of the paper could be increased by extending Figure 8 model to include the main remodelers performing ‘tug-of-war’ at promoters – RSC, SWI/SNF, INO80, ISW2, and have revised the model and text [P24L523 ~ P25L550] accordingly. (CHD1 and ISW1 act substantially on gene bodies and thus are not included in the revised model). As requested, we have also relabeled the figure to distinguish between ATP-binding and ATP-hydrolysis at the steps of 1D diffusion and nucleosome remodeling/remodeler dissociation. We thank the reviewer for suggesting these changes, which improves the model’s general accessibility.

2) The authors should comment on the perceived discrepancy in the relative abundance of the remodelers. The authors should consider adding the proposed quantification via Western Blot if feasible.

We appreciate the reviewer’s perceptive comment on the discrepancy between the in-gel JF646 intensities as shown in Figure 1—figure supplement 1A and the # molecules/cell presented by Ho et al., 2018, shown in Figure 7A. The discrepancy can be attributed to the protein extraction method (Amberg et al., 2006) we used which does not provide quantitative protein extraction due to the yeast cell wall barrier. For example, SDS-PAGE in-gel JF646 analysis of RSC lysates shows substantial signal retained in the insoluble pellet in addition to the supernatant (data presented in revised Figure 1—figure supplement 1A). Even with pellet analysis, we are unsure that there are no additional losses incurred during extraction. The purpose of our original presentation was simply to show the integrity of the HaloTagged factors at their expected sizes. We rely on Ho et al.’s extensive meta-analysis of twenty-one yeast proteomic studies for the best available quantitative estimate of remodeler abundance, with the assumption, now explicitly stated in the revised text [P18L371], that the functional fusions are similar in abundance to the untagged proteins.

As a rough test for consistency with Ho et al., we quantified the live-cell JF intensities by measuring the initial nuclear fluorescence upon laser illumination in our fast-tracking videos. We found that INO80 and ISW2 abundances are substantially lower than RSC but similar to each other (Author response image 1), comparable to the similar median values presented by Ho et al. (INO80: 2976 ± 661; ISW2: 2705 ± 316). (In the revised table Figure 7A, we substitute the average N_molecules_ values for the median, which more accurately represents the central value when datasets have substantial outliers).

**Author response image 1. sa2fig1:** 

3) The authors should comment on several important technical points raised by the reviewers, in particular in terms of the imaging conditions (single remodelers rather than clusters, potential effect of exposure to laser light, media flow during data acquisition) and data analysis (decomposition of histograms in Figures 1 and 2. The authors should also comment on the cell cycle phase and explicitly describe if/how it was taken into account.Appropriate textual changes to the manuscript would improve clarity on these points.

We appreciate the technical issues raised and clarifications requested by reviewer 3 and the other reviewers. Please see detailed response to specific points below, which have been incorporated into the text and supplemental figures.

Reviewer #1 (Recommendations for the authors):1) It appears to this reviewer that the Log D and jump distance histograms in Figures 1 and 2 should be decomposed into three components: chromatin-bound, intermediate diffusive, and free diffusive, since the authors have identified an intermediate diffusive population that can transition between bound and free states, which are shown in Figure 3.

Thank you for the recommendation. We understand that Figures 1 and 2 LogD histograms may confuse readers by our presentation of 2-component fits when in fact there is an intermediate population described in Figure 3. Hence, we revised our original figures 1 and 2 to show the Spot-On values for fraction bound, and moved the LogD histograms to Figure 1—figure supplement 4. The vbSPT analysis shown in Figure 3 reveals that in addition to two main diffusive populations there is a third population composed of trajectories containing both bound and free states.

2) The authors use the number of molecules of remodelers from published results (Ho, 2018) to calculate target-search kinetics. It appears to this reviewer that the published data does not match the data in Figure S1A. For instance, Figure 7A shows that INO80 is more abundance than ISW2; however, Figure S1A shows the opposite results. Given that these remodelers have been endogenously tagged, it would be possible to quantify the number of molecules of remodelers through western blotting by comparing to known concentrations of HaloTag.

Please see above response to Editor’s Essential revisions.

3) P27L603: It should be 10 ms exposure time.

We have corrected the text accordingly [P30L654].

4) P27L605: Please explain why each video lasts 1.5 min.

We have added this paragraph to methods for clarification [P30L656]:

Single-molecule imaging was performed using dSTORM (direct stochastic optical reconstruction microscopy) (Heilemann et al., 2008; Rust, Bates, and Zhuang, 2006). Each video begins with global nuclear fluorescence on laser excitation of essentially all JF552-labelled molecules before photoconversion to a dark, non-fluorescent state. The duration of the initial excitation which pushes fluorescent JF552 to the dark state is ~10 s, depending on the density of labeled Halo-proteins, which is a function of protein expression and dye labeling concentration adjusted for the specific protein. Once in the dark state, JF552 spontaneously and stochastically converts back to the fluorescent state (Grimm et al., 2015; Zheng et al., 2019).This conversion, a characteristic of fluorophore photochemistry, is sufficiently slow such that mostly one JF552 molecule per yeast nucleus is fluorescent at any given time during data acquisition, allowing for an unambiguous spatio-temporal record of single JF552-labeled proteins. To ensure that we can harvest a ‘substack’ of at least 5000 frames (10 ms/frame) (See Single molecule image analysis), we routinely record videos of ~1.5 min.

5) P27L607: It should be 250 ms exposure time.

We have corrected the text accordingly, thank you for catching the error [P31L676].

6) P27L612: Please explain again why each video lasts 7-9 min.

Upon 555 nm laser illumination, we record ~ 5 min [correction of 7-9 min] videos starting with fluorescent JF552 pushed to the dark state for > 30 s, followed by single-molecule imaging of at least 750 frames at 250 ms/frame. This correction is included in revised methods [P31L681].

7) P28L638: Please check the equation.D=12d×MSD(dt)dt8) P32L716: Please check the reference.

The reference is now in press, and we have changed accordingly [P25L553, P31L673, P35L790, P36L824].

Reviewer #2 (Recommendations for the authors):The model presented in figure 8 restricts itself to two of the remodelers (RSC and INO80) addressed, and does not reflect the full scope of the results obtained in this work. It would be helpful if changes could be made to the model figure in order to embrace all six remodelers studied, highlighting their kinetics similarities, and the resulting tug-of-war at promoters. Notably, it would be helpful to consolidate and clarify the ATPase state, as ATP and ADP+Pi are written concomitantly at all stages of each remodeler presented. Moreover, the model would benefit from a more clear depiction of ATP binding in the enhancement of nucleosome browsing, and of ATP hydrolysis in the enhancement of dissociation.

Thank you for the comment. Please see detailed response to Essential revisions above.

Reviewer #3 (Recommendations for the authors):The paper is a technical tour-de-force, but I have a number of technical questions that need clarification. These arise partially from my ignorance, but also from the fact that there seem to be details missing in the methods section. Given that the paper should be read by a maximal number of chromatin biologists, I think it would greatly enhance the impact of the paper to include explanations concerning the following points, either in the methods section or in the text itself.1. How do they know that they are actually imaging SINGLE chromatin remodelers in every case, and not clusters of a given remodeler, for example, bound at multiple nucleosomes at a promoter (or cluster of promoters) ? I understand that they use a pulsed laser to activate the Halo tag fluorochrome, but it seems to me that they need to quantify the number of molecules each signal might represent. Hansen and Darzacq have worked out means to do this. The authors could also estimate the number of total molecules of each remodeler this way, which would be useful to know, especially to see if this influences occupancy rates.

Please see our response to Reviewer 1’s Comment #4 above. In addition, to further illustrate the point, we refer the reviewer to a sequence of frames in Author response image 2, which shows one-step spot appearance (conversion from dark to fluorescent state, t=78.81 s) and one-step disappearance (photobleaching, t=79.50 s). In addition, we customize tracking parameters (See Methods) to filter out rare instances of overlapping spots.

Regarding the number of molecules per cell, see our response to Essential revisions above.

2. The pixel size (over 100 nm) is quite large – certainly large enough to contain multiple protein complexes. How large are the signals they monitor ? Did the try to reduce pixel size (it is possible with the LSM 710 and 100x objective)?

Pixel size is a fixed number based on camera resolution and microscope objective. Despite the 107 nm square pixel size and the spread of signal over several pixels, the ability to separately image individual molecules allows localization of the centroid by Gaussian fitting the signal distribution over multiple pixels to the center of the PSF (point spread function) with a precision of ~30 nm (Thompson, Larson, and Webb, 2002; Yildiz et al., 2003). [P31L693]

3. The authors use an imaging scheme worked out by Hansen et al. for "rapid" and "slower" movements. In one they image at 10ms intervals for 1.5 min, but for the slower one they image at 250ms intervals for 7-9 minutes. This latter scheme implies a lot of exposure to laser light. Did they check that there was no impact on cell division rate after the slower, longer imaging protocol ? It is a massive dose, and the time needed for cell division should be monitored under these conditions.

We share the reviewer’s concern about photodamage. For fast tracking, we excite the sample with ~1 kW/cm^2^ continuous laser and image a 128x128 pixel field of view (containing 4-6 yeast cells) for ~1.5 minutes at 10 millisecond camera integration time. Under these conditions, illuminated cells, monitored every 30 minutes for up to 3 hours, exhibited no observable cellular damage and underwent bud growth and cell division similar to unexposed cells. This finding is provided by Nguyen et al., 2021 from our lab (Molecular Cell, in press) and re-stated in our methods with the citation for the readers [P31L671]. (Note that the use of 555 nm laser excitation for JF552, or longer excitation wavelengths in the red and near IR is crucial; a similar exposure to blue or violet laser light is deleterious to yeast). As a further check, we separately analyzed the first and second halves of our fast-tracking videos and found no substantial differences, indicating that there are little or no progressive changes in diffusive behaviors that we can detect over 1.5 min [P31L673]. This is now documented in Figure 1—figure supplement 3B. Importantly, for slow-tracking, low laser illumination requiring only 5% of the previous laser power [P31L677] provides sufficient signal over the ~5 min of imaging. These details are given in methods.

4. The authors say that they image for 2 h after the immobilization of cells on a cover slide. Are the cells continually bathed in medium during imaging ? this should be stated, if so. During 2 hours without glucose yeast rapidly deplete their ATP levels and this strongly influences chromatin mobility and remodeler activity (as shown 20 years ago, but also in this paper by walker motif mutation). It leads one to wonder if the slower moving remodeler tracks were taken at the end of the 2h period and the faster ones at the early time points ? Or did they keep a flow of fresh media onto the cells ? This is a very important technical point that should be explained in the methods.

We share the concern over general yeast health during imaging. After harvesting log-phase cells and 3x washing with Synthetic Complete Medium (with glucose) to remove excess dye, yeasts are immobilized on a coverslip for microscopy (in imaging chamber Attofluor Cell Chamber A7816). After immobilization and additional washing, ~1.5 mL of fresh medium is added to the chamber in which the cells are bathed continually for the duration of the session (~2 h). We include these details in the Methods section [P29L619, P29L632]. Furthermore, comparison of results from the first and second halves of an imaging session shows no substantial differences, suggesting that cells remain in good health (presented in Figure 1—figure supplement 3A) [P30L634].

5. For the longer (but not shorter) tracking and subsequent calculation of diffusion coefficients the authors should be conscious that nuclear movement (or cellular and instrumental drift) can alter the track, if not normalized out. I guess the fact that they subtract H2B movement does normalize for cellular or instrumental drift, but perhaps they should state that. Other means exist, e.g. to track the center of the nucleus interpolated from a ring of nuclear pores or background nuclear fluorescence – and subtract that movement from the movement of the SPT. Some discussion of removal of such background drift should be mentioned in methods, or if this has already been ruled out elsewhere, the reference should be cited.

For stably-bound remodelers measured by slow-tracking, we present uncorrected, *apparent* diffusion coefficients, but provide H2B as a reference standard for global chromatin and nuclear motions and instrument drift. We added a statement in the text to be clear about this [P15L317].

6. Finally, the authors assume that the nucleosome remodelers are almost exclusively involved in transcription, and based on this they make the tug-of-war argument that INO80 and ISWI2 are counteracting RSC and Swi/SNF action. However, the two remodelers that have slightly different diffusion histograms (INO80 and ISWI2) are both implicated in DNA replication (and also in DNA repair), as well as transcription. The Tsukiyama lab has clearly demonstrated their roles in replication, and various authors provided convincing evidence that INO80 is important for paused fork restart and other repair events. DNA repair may be too rare an event (compared to transcription) to bias the movement observed, but replication could definitely bias the mobility/diffusion or Spot-on profiles, if a fraction of the cells imaged were in S phase.This raises a number of questions:a) Were the cells imaged exclusively in G1 phase ? That is, were only cells without buds monitored ? This should be stated if so.

We imaged asynchronous log phase cells without selecting specific stages of the cell cycle. Thus, cells undergoing S-phase are included in our analysis, and the measured dynamics would include remodeler participation in replication as well as transcription. We thank the reviewer for the reminder and have revised the text accordingly [P5L144] and [P30L652].

b) If not, did they monitor INO80 or ISWi2 diffusion specifically in S phase cells and did this lead to the same or a different profile of chromatin-bound/-free fractions and diffusion co-efficients?

We have not selected or synchronized for cells in S-phase in this study, and plan to investigate the interesting question of remodeler participation in replication and repair, particularly for INO80, in the near future.

c) The diffusion histograms are fit to two Gaussian curves, but they would also be compatible with the superposition of multiple populations. Given that INO80 (and probably ISWI2) respond to phosphorylation, and may be regulated by cell cycle phase – it might be that there are more than simply a "bound" and an "unbound" population. This should – at the very least be discussed.I think it should be mentioned that these specific two remodelers are involved in replication – and what that might imply – unless all cells imaged were in G1 phase. If they were all G1 phase cells, then it should have been mentioned (sorry if I missed it).d) There could also be multiple variant complexes, containing additional subunits or being post-translationally modified, that might account for the complexity of Diffusion coefficients. This should at least be mentioned somewhere in the discussion.

Response to comments (c and d):

We thank the reviewer for the important reminder that remodelers are not exclusive to transcription, and have revised the text to include this in [P4L96], [P19L390], and [P20L421]. Indeed, INO80 and ISW2 are prime candidates for dynamic analysis under specific conditions of replication and repair and will be subjects of future analysis. We also agree that there could be various sub-populations within our two observable diffusive populations which are beyond dissection by SMT. We have included this in the Results [P8L186] and Discussion [P26L573].